# ON THE FOUNDATIONS OF SHORTCUT LEARNING

**Katherine L. Hermann**[1]**, Hossein Mobahi**[2]**, Thomas Fel**[1]**, and Michael C. Mozer**[1]

Google {[1]DeepMind, [2]Research}, Mountain View, CA, USA
`{hermannk, hmobahi, thomasfel, mcmozer}@google.com`

## ABSTRACT

Deep-learning models can extract a rich assortment of features from data. Which features a model uses depends not only on *predictivity*—how reliably a feature indicates training-set labels—but also on *availability*—how easily the feature can be extracted from inputs. The literature on shortcut learning has noted examples in which models privilege one feature over another, for example texture over shape and image backgrounds over foreground objects. Here, we test hypotheses about which input properties are more available to a model, and systematically study how predictivity and availability interact to shape models' feature use. We construct a minimal, explicit generative framework for synthesizing classification datasets with two latent features that vary in predictivity and in factors we hypothesize to relate to availability, and we quantify a model's shortcut bias—its over-reliance on the shortcut (more available, less predictive) feature at the expense of the core (less available, more predictive) feature. We find that linear models are relatively unbiased, but introducing a single hidden layer with ReLU or Tanh units yields a bias. Our empirical findings are consistent with a theoretical account based on Neural Tangent Kernels. Finally, we study how models used in practice trade off predictivity and availability in naturalistic datasets, discovering availability manipulations which increase models' degree of shortcut bias. Taken together, these findings suggest that the propensity to learn shortcut features is a fundamental characteristic of deep nonlinear architectures warranting systematic study given its role in shaping how models solve tasks.

## 1 INTRODUCTION

Natural data domains provide a rich, high-dimensional input from which deep-learning models can extract a variety of candidate features. During training, models determine which features to rely on. Following training, the chosen features determine how models generalize. A challenge for machine learning arises when models come to rely on *spurious* or *shortcut* features instead of the core or defining features of a domain (Arjovsky et al., 2019; McCoy et al., 2019; Geirhos et al., 2020; Singla & Feizi, 2022). Shortcut "cheat" features, which are correlated with core "true" features in the training set, obtain good performance on the training set as well as on an iid test set, but poor generalization on out-of-distribution inputs. For instance, ImageNet-trained CNNs classify primarily according to an object's texture (Baker et al., 2018; Geirhos et al., 2018a; Hermann et al., 2020), whereas people define and classify solid objects by shape (e.g., Landau et al., 1988). Focusing on texture leads to reliable classification on many images but might result in misclassification of, say, a hairless cat, which has wrinkly skin more like that of an elephant. The terms "spurious" and "shortcut" are largely synonymous in the literature, although the former often refers to features that arise unintentionally in a poorly constructed dataset, and the latter to features easily latched onto by a model. In addition to a preference for texture over shape, other common shortcut features include a propensity to classify based on image backgrounds rather than foreground objects (Beery et al., 2018; Sagawa et al., 2020a; Xiao et al., 2020; Moayeri et al., 2022), or based on individual diagnostic pixels rather than higher-order image content (Malhotra et al., 2020).

The literature examining feature use has often focused on *predictivity*—how well a feature indicates the target output. Anomalies have been identified in which networks come to rely systematically on one feature over another when the features are equally predictive, or even when the preferred feature has lower predictivity than the non-preferred feature (Beery et al., 2018; Pérez et al., 2018; Tachet

et al., 2018; Arjovsky et al., 2019; McCoy et al., 2019; Hermann & Lampinen, 2020; Shah et al., 2020; Nagarajan et al., 2020; Pezeshki et al., 2021; Fel et al., 2023). Although we lack a general understanding of the cause of such preference anomalies, several specific cases have been identified. For example, features that are linearly related to classification labels are preferred by models over features that require nonlinear transforms (Hermann & Lampinen, 2020; Shah et al., 2020). Another factor leading to anomalous feature preferences is the redundancy of representation, e.g., the size of the pixel footprint in an image (Sagawa et al., 2020a; Wolff & Wolff, 2022; Tartaglini et al., 2022).

Because predictivity alone is insufficient to explain feature reliance, here we explicitly introduce the notion of *availability* to refer to the factors that influence the likelihood that a model will use a feature more so than a purely statistical account would predict. A more-available feature is easier for the model to extract and leverage. Past research has systemtically manipulated predictivity; in the present work, we systematically manipulate both predictivity *and* availability to better understand their interaction and to characterize conditions giving rise to shortcut learning. Our contributions are:

- We define quantitative measures of predictivity and availability using a generative framework that allows us to synthesize classification datasets with latent features having specified predictivity and availability. We introduce two notions of availability relating to singular values and nonlinearity of the data generating process, and we quantify shortcut bias in terms of how a learned classifier deviates from an optimal classifier in its feature reliance.

- We perform parametric studies of latent-feature predictivity and availability, and examine the sensitivity of different model architectures to shortcut bias, finding that it is greater for nonlinear models than linear models, and that model depth amplifies bias.

- We present a theoretical account based on Neural Tangent Kernels (Jacot et al., 2018) which indicates that shortcut bias is an inevitable consequence of nonlinear architectures.

- We show that vision architectures used in practice can be sensitive to non-core features beyond their predictive value, and show a set of availability manipulations of naturalistic images which push around models' feature reliance.

## 2 RELATED WORK

The propensity of models to learn spurious (Arjovsky et al., 2019) or shortcut (Geirhos et al., 2020) features arises in a variety of domains (Heuer et al., 2016; Gururangan et al., 2018; McCoy et al., 2019; Sagawa et al., 2020a) and is of interest from both a scientific and practical perspective. Existing work has sought to understand the extent to which this tendency derives from the statistics of the training data versus from model inductive bias (Neyshabur et al., 2014; Tachet et al., 2018; Pérez et al., 2018; Rahaman et al., 2019; Arora et al., 2019; Geirhos et al., 2020; Sagawa et al., 2020b;a; Pezeshki et al., 2021; Nagarajan et al., 2021), for example a bias to learn simple functions (Tachet et al., 2018; Pérez et al., 2018; Rahaman et al., 2019; Arora et al., 2019). Hermann & Lampinen (2020) found that models preferentially represent one of a pair of equally predictive image features, typically whichever feature had been most linearly decodable from the model at initialization. They also identified cases where models relied on a less-predictive feature that had a linear relationship to task labels over a more-predictive feature that had a nonlinear relationship to labels. Together, these findings suggest that predictivity alone is not the only factor that determines model representations and behavior. A theoretical account by Pezeshki et al. (2021) studied a situation in supervised learning in which minimizing cross-entropy loss captures only a subset of predictive features, while other relevant features go unnoticed. They introduced a formal notion of *strength* that determines which features are likely to dominate a solution. Their notion of strength confounds predictivity and availability, the two concepts which we aim to disengangle in the present work.

Work in the vision domain has studied which features vision models rely on when trained on natural image datasets. For example, ImageNet models prefer to classify based on texture rather than shape (Baker et al., 2018; Geirhos et al., 2018a; Hermann et al., 2020) and local rather than global image content (Baker et al., 2020; Malhotra et al., 2020), marking a difference from how people classify images (Landau et al., 1988; Baker et al., 2018). Other studies have found that image backgrounds play an outsize role in driving model predictions (Beery et al., 2018; Sagawa et al., 2020a; Xiao et al., 2020). Two studies manipulated the quantity of a feature in an image to test how this changed model behavior. Tartaglini et al. (2022) probed pretrained models with images containing a shape consistent

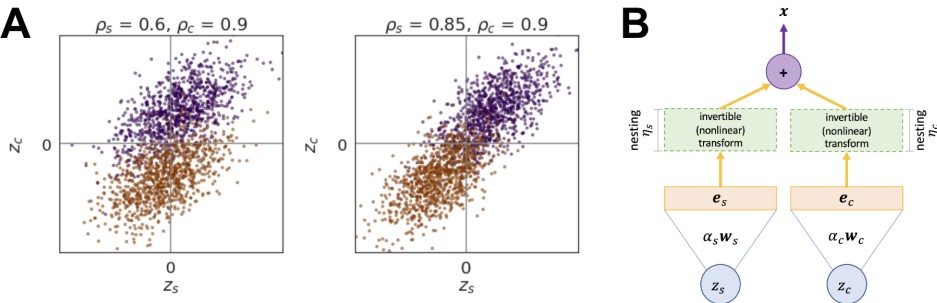

Figure 1: **Synthetic data.** A: Two datasets differing in the predictivity of $z_s$. B: Schematic of the embedding procedure manipulating availability via the mapping from $z$ to $x$. Dashed boxes are optional.

with one class label and a texture consistent with another, where the texture was present throughout the image, including in the background. They varied the opacity of the background, and as the the texture became less salient, models increasingly classified by shape. Wolff & Wolff (2022) found that when images had objects with opposing labels, models preferred to classify by the object with the larger pixel footprint, an idea we return to in Section 5. The development of methods for reducing bias for particular features over others is an active area (e.g. Arjovsky et al., 2019; Geirhos et al., 2018a; Hermann et al., 2020; Robinson et al., 2021; Minderer et al., 2020; Sagawa et al., 2020b; Ryali et al., 2021; Kirichenko et al., 2022; Teney et al., 2022; Tiwari & Shenoy, 2023; Ahmadian & Lindsten, 2023; Pezeshki et al., 2021; Puli et al., 2023; LaBonte et al., 2023), important for improving generalization and addressing fairness concerns (Zhao et al., 2017; Buolamwini & Gebru, 2018).

## 3  GENERATIVE PROCEDURE FOR SYNTHETIC DATASETS

To systematically explore the role of predictivity and availability, we construct synthetic datasets from a generative procedure that maps a pair of latent features, $z = (z_s, z_c)$, to an input vector $x \in \mathbb{R}^d$ and class label $y \in \{-1, +1\}$. The subscripts $s$ and $c$ denote the latent dimensions that will be treated as the potential *shortcut* and *core* feature, respectively. The procedure draws $z$ from a multivariate Gaussian conditioned on class,

$$z \,|\, y \sim \mathcal{N}\left(\begin{bmatrix} y\,\mu_s \\ y\,\mu_c \end{bmatrix}, \begin{bmatrix} 1 & \sigma_{sc} \\ \sigma_{sc} & 1 \end{bmatrix}\right),$$

with $|\sigma_{sc}| < 1$. Through symmetry, the optimal decision boundary for latent feature $i$ is $z_i = 0$, allowing us to define the feature predictivity $\rho_i \equiv \Pr(y = \text{sign}(z_i))$. For Gaussian likelihoods, this predictivity is achieved by setting $\mu_i = \sqrt{2}\,\text{erf}^{-1}(2\rho_i - 1)$. Figure 1A shows sample latent-feature distributions for $\rho_c = 0.9$ and two levels of $\rho_s$.

**Availability manipulations**. Given a latent vector $z$, we manipulate the hypothesized availability (hereafter, simply *availability*) of each feature independently through an embedding procedure, sketched in Figure 1B, that yields an input $x \in \mathbb{R}^d$. We posit two factors that influence availability of feature $i$, its *amplification* $\alpha_i$ and its *nesting* $\eta_i$. Amplification $\alpha_i$ is a scaling factor on an embedding, $e_i = \alpha_i w_i z_i$, where $w_i \in \mathbb{R}^d$ is a feature-specific random unit vector. Amplification includes manipulations of redundancy (replicating a feature) and magnitude (increasing a feature's dynamic range). Nesting $\eta_i$ is a factor that determines ease of recovery of a latent feature from an embedding. We assume a nonlinear, rank-preserving transform, $e_i' = f_{\eta_i}(e)$, where $f_{\eta_i}$ is a fully connected, random net with $\eta_i \in \mathbb{N}$ tanh layers in cascade. For $\eta_i = 0$, feature $i$ remains in explicit form, $e_i' = e_i$; for $\eta_i > 0$, the feature is recoverable through an inversion of increasing complexity with $\eta_i$. To complete the data generative process, we combine embeddings by summation: $x = e_s' + e_c'$.

**Assessing shortcut bias**. Given a synthetic dataset and a model trained on this dataset, we assess the model's *reliance* on the shortcut feature, i.e., the extent to which the model uses this feature as a basis for classification decisions. When $z_c$ and $z_s$ are correlated ($\sigma_{sc} > 0$), some degree of reliance on the shortcut feature is Bayes optimal (see orange dashed line in Figure 6B). Consequently, we need to assess reliance relative to that of an optimal classifier. We perform this assessment in latent space, where the Bayes optimal classifier can be found by linear discriminant analysis (LDA) (see

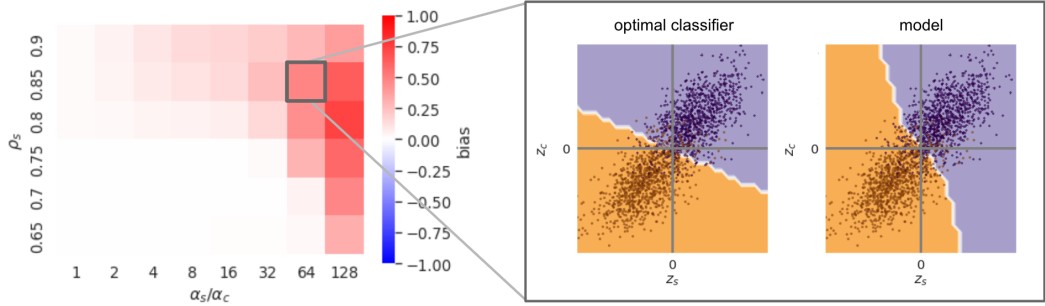

Figure 2: **Deep nonlinear models can prefer a less-predictive but more-available feature to a more-predictive but less-available one.** The color of each cell in the heatmap indicates the mean bias of a model as a function of the availability and predictivity of the shortcut feature, $z_s$. The inset shows in faint coloring the decision surface for an optimal Bayes classifier (LDA) and a trained model. Overlaid points are a subset of training instances. The model obtains a shortcut bias of $0.53$.

Figure 2 inset). The *shortcut bias* is the reliance of a model on the shortcut feature over that of the optimal, LDA:

$$bias = reliance_{\text{model}} - reliance_{\text{optimal}}.$$

Appendix C.1 describes and justifies our reliance measure in detail. For a given model $\mathcal{M}$, whether trained net or LDA, we probe over the latent space and determine for each probe $\boldsymbol{z}$ the binary classification decision, $\hat{y}_{\mathcal{M}(\boldsymbol{z})}$ (see Figure 2). Shortcut reliance is the difference between the model's alignment (covariance) with decision boundaries based only on the shortcut and core features:

$$reliance_{\mathcal{M}} = \frac{2}{n}\left(\left|\sum_i \hat{y}_{\mathcal{M}_i}\text{sign}(z_{s_i})\right| - \left|\sum_i \hat{y}_{\mathcal{M}_i}\text{sign}(z_{c_i})\right|\right),$$

where $n$ is the number of probe items. Both the reliance score and shortcut bias are in $[-1, +1]$.

## 4 EXPERIMENTS MANIPULATING FEATURE PREDICTIVITY AND AVAILABILITY

**Methodology**. Using the procedure described in Section 3, we conduct controlled experiments examining how feature availability and predictivity and model architecture affect the shortcut bias. We sample class-balanced datasets with 3200 train instances, 1000 validation instances, and 900 probe (evaluation) instances that uniformly cover the $(z_s, z_c)$ space by taking a Cartesian product of 30 $z_s$ evenly spaced in $[-3\mu_s, +3\mu_s]$ and 30 $z_c$ evenly spaced in $[-3\mu_c, +3\mu_c]$. In all simulations, we set $d = 100$, $\eta_c = \eta_s = 0$, $\rho_c = 0.9$, $\sigma_{sc} = 0.6$. We manipulate shortcut-feature predictivity with the constraint that it is lower than core-feature predictivity but still predictive, i.e., $0.5 < \rho_s < \rho_c = 0.9$. Because only the relative amplification of features matters, we vary the ratio $\alpha_s/\alpha_c$, with the shortcut feature more available, i.e., $\alpha_s/\alpha_c \geq 1$. We report the means across 10 runs (Appendix C.3).

**Models prefer the more available feature, even when it is less predictive**. We first test whether models prefer $z_s$ when it is more available than $z_c$, including when it is less predictive, while holding model architecture fixed (see Appendix C.3). Figure 2 shows that when predictivity of the two features is matched ($\rho_c = \rho_s = 0.9$), models prefer the more-available feature. And given sufficiently high availability, models can prefer the less-predictive but more-available shortcut feature to the more-predictive but less-available core feature. *Availability can override predictivity.*

**Model depth increases shortcut bias.** In the previous experiment, we used a fixed model architecture. Here, we investigate how model depth and width influence shortcut bias when trained with $\alpha_s/\alpha_c = 64$ and $\rho_s = 0.85$, previously shown to induce a bias (see Figure 2, gray square). As shown in Figure 3A, we find that bias increases as a function of model depth when dataset is held fixed.

**Model nonlinearity increases shortcut bias.** To understand the role of the hidden-layer activation function, we compare models with linear, ReLU, and Tanh activations while holding weight initialization and data fixed. As indicated in Figure 3B, nonlinear activation functions induce a larger shortcut bias than their linear counterpart.

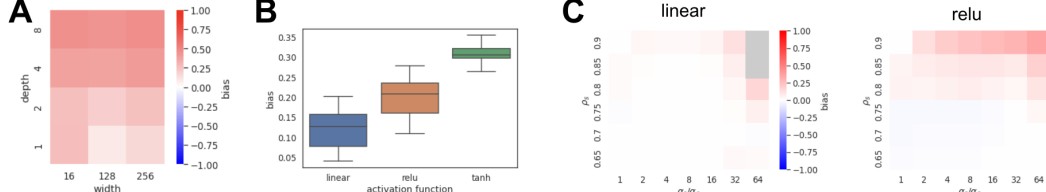

Figure 3: A: **Model depth increases shortcut bias.** The color of each cell indicates the mean bias of an MLP with ReLU hidden activation-functions, for various model widths and depths, trained on data with a shortcut feature that is more available ($\alpha_s/\alpha_c = 64$) but less predictive ($\rho_s = 0.85$) than the core feature. **Model nonlinearity increases shortcut bias.** B: Shortcut bias for three hidden activation functions for a deep MLP with width 128 and depth 2, trained on datasets where predictivity is matched ($\rho_s = \rho_c = 0.9$), but shortcut availability is higher ($\alpha_s/\alpha_c = 32$). A shortcut bias is more pronounced when the model contains a nonlinear activation function. C: Shortcut bias for MLPs with a single hidden layer and a hidden activation function that is either linear (left) or ReLU (right), for various shortcut feature availabilities ($\alpha_s/\alpha_c$) and predictivities ($\rho_s$). See B.1 for Tanh.

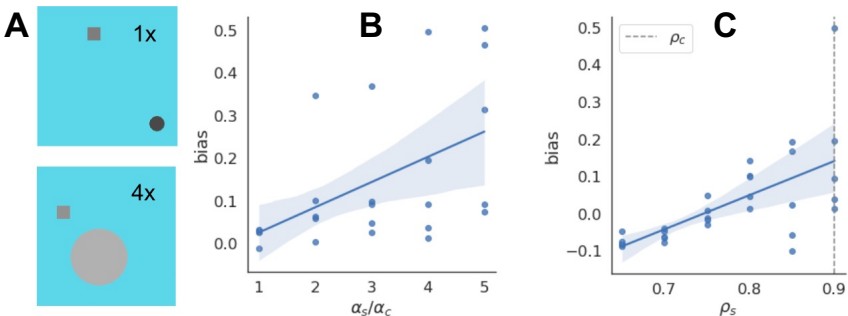

Figure 4: **ResNet-18 prefers a shortcut feature when availability is instantiated as the pixel footprint of an object (feature), even when that feature is less predictive.** A: Sample images. B: Shortcut bias increases as a function of relative availability of the shortcut feature when features are equally predictive ($\rho_s = \rho_c = 0.9$), consistent with Wolff & Wolff (2022). C: Even when the shortcut feature is less predictive, models have a shortcut bias due to availability, when $\alpha_s/\alpha_c = 4$.

**Feature nesting increases shortcut bias.** The synthetic experiments reported above all manipulate availability with the amplitude ratio $\alpha_s/\alpha_c$. We also conducted experiments manipulating a second factor we expected would affect availability (Hermann & Lampinen, 2020; Shah et al., 2020), the relative nesting of representations, i.e., $\eta_c - \eta_s \geq 1$. We report these experiments in Appendix C.3.

## 5  AVAILABILITY MANIPULATIONS WITH SYNTHETIC IMAGES

What if we instantiate the same latent feature space studied in the previous section in images? We form shortcut features that are analogous to texture or image background—features previously noted to preferentially drive the behavior of vision models (e.g. Geirhos et al., 2018a; Baker et al., 2018; Hermann et al., 2020; Beery et al., 2018; Sagawa et al., 2020a; Xiao et al., 2020). Building on the work of Wolff & Wolff (2022), we hypothesize that these features are more available because they have a large footprint in an image, and hence, by our notions of availability, a large $\alpha_s$.

**Methods**. We instantiate a latent vector $z$ from our data-generation procedure as an image. Each feature becomes an object ($z_s$ a circle, $z_c$ a square) whose color is determined by the respective feature value. Following Wolff & Wolff (2022), we manipulate the availability of each feature in terms of its size, or pixel footprint. We randomly position the circle and square entirely within the image, avoiding overlap, yielding a $224 \times 224$ pixel image (Figure 4A, Appendix C.4).

**Results**. Figure 4B presents the shortcut bias in ResNet18 as a function of shortcut-feature availability (footprint) when the two features are equally predictive ($\rho_s = \rho_c = 0.9$). In Figure 4C, the availability

ratio is fixed at $\alpha_s/\alpha_c = 4$, and the shortcut bias is assessed as a function of $\rho_s$. ResNet18 is biased toward the more available shortcut feature even when it is less predictive than the core feature. Together, these results suggest that a simple characteristic of image contents—the pixel footprint of an object—can bias models' output behavior, and may therefore explain why models can fail to leverage typically-smaller foreground objects in favor of image backgrounds (Section 7).

## 6 THEORETICAL ACCOUNT

In our empirical investigations, we quantified the extent to which a trained model deviates from a statistically optimal classifier in its reliance on the more-available feature using a measure which considered the basis for probe-instance classifications. Here, we use an alternative approach of studying the sensitivity of a Neural Tangent Kernel (NTK) (Jacot et al., 2018) to the availability of a feature. The resulting form presents a crisp perspective of how predictivity and availability interact. In particular, we prove that availability bias is absent in linear networks but present in ReLU networks. The proof for the theorems of this section is given in the Supplementary Materials.

For tractability of the analysis, we consider a few simplifying assumptions. We focus on 2-layer fully connected architectures for which the kernel of the ReLU networks admits a simple closed form. In addition, to be able to compute integrations that arise in the analysis, we resort to an asymptotic approximation which assumes covariance matrix is small. Specifically, we represent the covariance matrix as $s\,[\,1\,,\,\sigma_{12}\,;\,\sigma_{12}\,,\,1\,]$, where the scale parameter $s > 0$ is considered to be small. Finally, in order to handle the analysis for a ReLU kernel, we will use a polynomial approximation.

**Kernel Spectrum**. Consider a two-layer network with the first layer having a possibly nonlinear activation function. When the width of this model gets large, learning of this model can be approximated by a kernel regression problem, with a given kernel function $k(\,.\,,\,.\,)$ that depends on the architecture and the activation function. Given a distribution over the input data $p(\boldsymbol{x})$, we define a (linear) kernel operator as one that acts on a function $f$ to produce another function $g$ as in $g(\boldsymbol{x}) = \int_{\mathbb{R}^n} k(\boldsymbol{x}, \boldsymbol{z})\, f(\boldsymbol{z})\, p(\boldsymbol{z})\, d\boldsymbol{z}$. This allows us to define an eigenfunction $\phi$ of the kernel operator as one that satisfies,

$$\lambda\,\phi(\boldsymbol{x}) \;=\; \int_{\mathbb{R}^n} k(\boldsymbol{x}, \boldsymbol{z})\, \phi(\boldsymbol{z})\, p(\boldsymbol{z})\, d\boldsymbol{z}\,. \tag{1}$$

The value of $\lambda$ will be the eigenvalue of that eigenfunction when the eigenfunction $\phi$ is normalized as $\int_{\mathbb{R}^n} \phi^2(\boldsymbol{x})\, p(\boldsymbol{x})\, d\boldsymbol{x} \;=\; 1$.

**Form of** $p(\boldsymbol{z})$. Recall that in our generative dataset framework, we have a pair of latent features $z_c$ and $z_s$ that are embedded into a high dimensional space via $\boldsymbol{x}_{d\times 1} = \alpha_s z_s \boldsymbol{w}_s + \alpha_c z_c \boldsymbol{w}_c = \boldsymbol{U}_{d\times 2}\boldsymbol{A}_{2\times 2}\boldsymbol{z}_{2\times 1}$. With this expression, we switch terminology such that our $\boldsymbol{w}_i \to \boldsymbol{U}$ and $\alpha_i \to \boldsymbol{A}$, and therefore $\boldsymbol{A}$ is diagonal matrix with positive diagonal entries, and the columns of $\boldsymbol{U}$ are (approximately) orthonormal. Henceforth, we also refer to features with indices 1 and 2 instead of $s$ and $c$. An implication of orthonormal columns on $\boldsymbol{U}$ is that the dot product of any two input vectors $\boldsymbol{x}$ and $\boldsymbol{x}^{\dagger}$ will be independent of $\boldsymbol{U}$, i.e., $\langle \boldsymbol{x}, \boldsymbol{x}^{\dagger}\rangle = \langle \boldsymbol{A}\boldsymbol{z}, \boldsymbol{A}\boldsymbol{z}^{\dagger}\rangle$. Consequently, we can compute dot products in the original 2-dimensional space instead of in the $d$-dimensional embedding space. On the other hand, we will later see that the kernel function $k(\boldsymbol{x}_1, \boldsymbol{x}_2)$ of the two cases we study here (ReLU and linear) depends on their input only through the dot product $\langle \boldsymbol{x}_1, \boldsymbol{x}_2\rangle$ and norms $\|\boldsymbol{x}_1\|$ and $\|\boldsymbol{x}_2\|$ (self dot products). Thus, the kernel is entirely invariant to $\boldsymbol{U}$ and without loss of generality, we can consider the input to the model as $\boldsymbol{x} = \boldsymbol{A}\boldsymbol{z}$. Therefore,

$$\boldsymbol{x}^+ \sim \mathcal{N}\left(\begin{bmatrix} a_1\mu_1 \\ a_2\mu_2 \end{bmatrix}, \begin{bmatrix} a_1^2 & a_1 a_2 \sigma_{12} \\ a_1 a_2 \sigma_{12} & a_2^2 \end{bmatrix}\right),\; \boldsymbol{x}^- \sim \mathcal{N}\left(-\begin{bmatrix} a_1\mu_1 \\ a_2\mu_2 \end{bmatrix}, \begin{bmatrix} a_1^2 & a_1 a_2 \sigma_{12} \\ a_1 a_2 \sigma_{12} & a_2^2 \end{bmatrix}\right).$$

**Linear kernel function**. If the activation function is linear, then the kernel function simply becomes a standard dot product,

$$k(\boldsymbol{x}_1, \boldsymbol{x}_2) \;\triangleq\; \langle \boldsymbol{x}_1, \boldsymbol{x}_2\rangle\,. \tag{2}$$

The following theorem provides details about the spectrum of this kernel.

**Theorem 1** *Consider the kernel function $k(\boldsymbol{x}_1, \boldsymbol{x}_2) \triangleq \langle \boldsymbol{x}_1, \boldsymbol{x}_2\rangle$. The kernel operator associated with $k$ under the data distribution $p$ specified above has only one non-zero eigenvalue $\lambda = \|\boldsymbol{A}\boldsymbol{\mu}\|^2$ and its eigenfunction has the form $\phi(\boldsymbol{x}) = \frac{\langle \boldsymbol{A}\boldsymbol{\mu}, \boldsymbol{x}\rangle}{\|\boldsymbol{A}\boldsymbol{\mu}\|^2}$.*

**ReLU kernel function**. If the activation function is set to be a ReLU, then the kernel function is known to have the following form (Cho & Saul, 2009; Bietti & Bach, 2020):

$$k(\boldsymbol{x}_1, \boldsymbol{x}_2) \triangleq \|\boldsymbol{x}_1\| \|\boldsymbol{x}_2\| h\left(\left\langle \frac{\boldsymbol{x}_1}{\|\boldsymbol{x}_1\|}, \frac{\boldsymbol{x}_2}{\|\boldsymbol{x}_2\|} \right\rangle\right), \quad h(u) \triangleq \frac{1}{\pi}\left(u\left(\pi - \arccos(u)\right) + \sqrt{1-u^2}\right). \quad (3)$$

In order to obtain an analytical form for the eigenfunctions of the kernel under the considered data distribution, we resort to a quadratic approximation of $h$ by $\widehat{h}$ as $\widehat{h}(u) \triangleq \frac{815}{3072}(1+u)^2$. This approximation enjoys certain optimality criteria. Derivation details of this quadratic form are provided in the Supplementary Materials, as is a plot showing the quality of the approximation. We now focus on spectral characteristics of the approximate ReLU kernel. Replacing $h$ in the kernel function $k$ of Equation 3 with $\widehat{h}$, we obtain an alternative kernel function approximation for ReLUs:

$$k(\boldsymbol{x}, \boldsymbol{z}) \triangleq \|\boldsymbol{x}\| \|\boldsymbol{z}\| \widehat{h}\left(\left\langle \frac{\boldsymbol{x}}{\|\boldsymbol{x}\|}, \frac{\boldsymbol{z}}{\|\boldsymbol{z}\|} \right\rangle\right) = \|\boldsymbol{x}\| \|\boldsymbol{z}\| a^* \left(1 + \left\langle \frac{\boldsymbol{x}}{\|\boldsymbol{x}\|}, \frac{\boldsymbol{z}}{\|\boldsymbol{z}\|} \right\rangle\right)^2. \quad (4)$$

The following theorem characterizes the spectrum of this kernel.

**Theorem 2** *Consider the kernel function $k(\boldsymbol{x}, \boldsymbol{z}) \triangleq \|\boldsymbol{x}\| \|\boldsymbol{z}\| a^* \left(1 + \left\langle \frac{\boldsymbol{x}}{\|\boldsymbol{x}\|}, \frac{\boldsymbol{z}}{\|\boldsymbol{z}\|} \right\rangle\right)^2$. The kernel operator associated with $k$ under the data distribution $p$ specified above has only two non-zero eigenvalues $\lambda_1 = \lambda_2 = 2a^* \|\boldsymbol{A}\boldsymbol{\mu}\|^2$ with associated eigenfunctions given by*

$$\phi_1(\boldsymbol{x}) = \frac{\|\boldsymbol{x}\|}{\|\boldsymbol{A}\boldsymbol{\mu}\|} \frac{1 + \left\langle \frac{\boldsymbol{x}}{\|\boldsymbol{x}\|}, \frac{\boldsymbol{A}\boldsymbol{\mu}}{\|\boldsymbol{A}\boldsymbol{\mu}\|} \right\rangle^2}{2} \ \text{ and } \ \phi_2(\boldsymbol{x}) = \left\langle \frac{\boldsymbol{x}}{\|\boldsymbol{A}\boldsymbol{\mu}\|}, \frac{\boldsymbol{A}\boldsymbol{\mu}}{\|\boldsymbol{A}\boldsymbol{\mu}\|} \right\rangle.$$

## 6.1 SENSITIVITY ANALYSIS

We now assign a target value $y(\boldsymbol{x})$ to each input point $\boldsymbol{x}$. By expressing the kernel operator using its eigenfunctions, it is easy to show that $f(\boldsymbol{x}) = \sum_i (\phi_i(\boldsymbol{x}) \int_{\mathbb{R}^n} \phi_i(\boldsymbol{z}) \, y(\boldsymbol{z}) \, p(\boldsymbol{z}) \, d\boldsymbol{z})$ where $i$ runs over non-zero eigenvalues $\lambda_i$ of the kernel operator (see also Appendix G). We now restrict our focus to a binary classification scenario, $y : \mathcal{X} \to \{0, 1\}$. More precisely, we replace $p$ with our Gaussian mixture and set $y(\boldsymbol{x})$ to 1 and 0 depending on whether $\boldsymbol{x}$ is drawn from the positive or negative component of the mixture.

$$f(\boldsymbol{x}) = \sum_i \left(\phi_i(\boldsymbol{x})\left(\tfrac{1}{2} \int_{\mathbb{R}^2} \phi_i(\boldsymbol{z}) \, (1) \, p^+(\boldsymbol{z}) \, d\boldsymbol{z} + \tfrac{1}{2} \int_{\mathbb{R}^2} \phi_i(\boldsymbol{z}) \, (0) \, p^-(\boldsymbol{z}) \, d\boldsymbol{z}\right)\right) = \tfrac{1}{2} \sum_i \phi_i(\boldsymbol{x}) \phi_i(\boldsymbol{A}\boldsymbol{\mu}),$$

where $p^+$ and $p^-$ denote class-conditional normal distributions. Now let us tweak the availability of each feature by a diagonal scaling matrix $\boldsymbol{B} \triangleq \mathrm{diag}(\boldsymbol{b})$ where $\boldsymbol{b} \triangleq [b_1, b_2]$. Denote the modified prediction function as $g_{\boldsymbol{B}}$. That is, $g_{\boldsymbol{B}}(\boldsymbol{x}) \triangleq \tfrac{1}{2} \sum_i \phi_i(\boldsymbol{x}) \phi_i(\boldsymbol{B}\boldsymbol{A}\boldsymbol{\mu})$. The alignment between $f$ and $g$ is their normalized dot product,

$$\gamma(\boldsymbol{b}) \triangleq \int_{\mathbb{R}^2} \frac{f(\boldsymbol{x})}{\sqrt{\int_{\mathbb{R}^2} f^2(\boldsymbol{t}) p(\boldsymbol{t}) \, d\boldsymbol{t}}} \frac{g_{\boldsymbol{B}}(\boldsymbol{x})}{\sqrt{\int_{\mathbb{R}^2} g_{\boldsymbol{B}}^2(\boldsymbol{t}) p(\boldsymbol{t}) \, d\boldsymbol{t}}} p(\boldsymbol{x}) \, d\boldsymbol{x}. \quad (5)$$

We define the sensitivity of the alignment to feature $i$ for $i = 1, 2$ as its $m$'th order derivative of $\gamma$ w.r.t. $b_i$ evaluated at $\boldsymbol{b} = \boldsymbol{1}$ (which leads to identity scale factor $\boldsymbol{B} = \boldsymbol{I}$),

$$\zeta_i \triangleq \left(\frac{\partial^m}{\partial b_i^m} \gamma\right)_{\boldsymbol{b}=\boldsymbol{1}} = \left(\frac{\partial^m}{\partial b_i^m} \int_{\mathbb{R}^2} \frac{\sum_i \phi_i(\boldsymbol{x}) \phi_i(\boldsymbol{A}\boldsymbol{\mu})}{\sqrt{\int_{\mathbb{R}^2} \left(\sum_i \phi_i(\boldsymbol{x}) \phi_i(\boldsymbol{A}\boldsymbol{\mu})\right)^2 dt}} \frac{\sum_i \phi_i(\boldsymbol{x}) \phi_i(\boldsymbol{B}\boldsymbol{A}\boldsymbol{\mu})}{\sqrt{\int_{\mathbb{R}^2} \left(\sum_i \phi_i(\boldsymbol{x}) \phi_i(\boldsymbol{B}\boldsymbol{A}\boldsymbol{\mu})\right)^2 dt}} d\boldsymbol{x}\right)_{\boldsymbol{b}=\boldsymbol{1}}.$$

$\zeta_i$ indicates how much the model relies on feature $i$ to make a prediction. In particular, if whenever feature $i$ is more available than feature $j$, i.e. $a_i > a_j$, we also see the model is more sensitive to feature $i$ than feature $j$, i.e. $|\zeta_i| > |\zeta_j|$, the model is biased toward the more-available feature. In addition, when $a_i < a_j$ but $|\zeta_i| > |\zeta_j|$, the bias is toward the less-available feature. One can express the presence of these biases more concisely as $\mathrm{sign}\left((|\zeta_1| - |\zeta_2|)(a_1 - a_2)\right)$, where values of $+1$ and $-1$ indicate bias toward more- and less- available features, respectively. To verify either of these two cases via sensitivity, we must choose the lowest order $m$ that yields a non-zero $|\zeta_1| - |\zeta_2|$. On the other hand, if $|\zeta_1| - |\zeta_2| = 0$ for any choice of $a_1$ and $a_2$, the model is unbiased to feature availability.

The following theorems now show that linear networks are unbiased to feature availability, while ReLU networks are biased toward more available features.

**Theorem 3** *In a linear network, $|\zeta_1| - |\zeta_2|$ is always zero for any choice of $m \geq 1$ and regardless of the values of $a_1$ and $a_2$.*

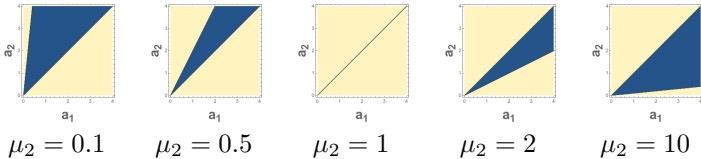

Figure 5: Plot of $\text{sign}((|\zeta_1| - |\zeta_2|)(a_1 - a_2))$ for ReLU network as a function of $a_1$ and $a_2$. We fix $\mu_1 = 1$ and vary $\mu_2 \in \{0.1, 0.5, 1, 2, 10\}$. Yellow and blue correspond to values $+1$ and $-1$ respectively.

**Theorem 4** *In a ReLU network,* $|\zeta_1| - |\zeta_2| = 0$ *for any* $1 \leq m \leq 8$. *The first non-zero* $|\zeta_1| - |\zeta_2|$ *happens at* $m = 9$ *and has the following form,*

$$|\zeta_1| - |\zeta_2| = \frac{5670}{\|A\mu\|^{18}} (a_1 a_2 \mu_1 \mu_2)^8 (a_1^2 \mu_1^2 - a_2^2 \mu_2^2). \tag{6}$$

A straightforward consequence of this theorem is that

$$\text{sign}\left((|\zeta_1| - |\zeta_2|)(a_1 - a_2)\right) = \text{sign}\left(\frac{5670}{\|A\mu\|^{18}} (a_1 a_2 \mu_1 \mu_2)^8 (a_1^2 \mu_1^2 - a_2^2 \mu_2^2)(a_1 - a_2)\right) = \text{sign}\left((a_1^2 \mu_1^2 - a_2^2 \mu_2^2)(a_1 - a_2)\right). \tag{7}$$

Recall from Section 3 that feature predictivity $\rho_i$ is related to $\mu_i$ via $\rho_i = \frac{1}{2}(1 + \text{erf}(\frac{\mu_i}{\sqrt{2}}))$. Observe that $\rho_i$ is an increasing function in $\mu_i$; thus, bigger $\mu_i$ implies larger $\rho_i$. That together with (7) provides a crisp interpretation of the trade-off between predictivity and availability. For example, when the latent features are equally predictive ($\mu_1 = \mu_2$), the sign becomes $+1$ for any (non-negative) choice of availability parameters $a_1$ and $a_2$. Thus, for equally predictive features, the ReLU networks are always biased toward the more available feature. Figure 5 shows some more examples with certain levels of predictivity. The coloring indicates the direction of the availability bias (only the boundaries between the blue and the yellow regions have no availability bias).

## 7 Feature availability in naturalistic datasets

We have seen that models trained on controlled, synthetic data are influenced by availability to learn shortcuts when a nonlinear activation function is present. How do feature predictivity and availability in naturalistic datasets interact to shape the behavior of models used in practice? To test this, we train ResNet18 models (He et al., 2016) to classify naturalistic images by a binary core feature irrespective of the value of a non-core feature. We construct two datasets by sampling images from Waterbirds (Sagawa et al., 2020a) (core: Bird, non-core: Background), and CelebA (Liu et al., 2015) (core: Attractive, non-core: Smiling). See C.5 for additional details.

**Sensitivity to the non-core feature beyond a statistical account**. Figures 6A and B show that, for both datasets, as the training-set predictivity of the non-core feature increases, model accuracy dramatically increases for congruent probes and decreases for incongruent ones. In contrast, a Bayes optimal classifier is far less sensitive to the predictivity of the non-core feature. Thus, models are more influenced by the non-core feature than what we would expected based solely on predictivity. This heightened sensitivity implies that models prioritize the non-core feature more than they should, given its predictive value. Thus, in the absence of predictivity as an explanatory factor, we conclude that the non-core feature is more *available* than the core feature.

**Availability manipulations**. Motivated by the result that Background is more available to models than the core Bird (Figure 6A), we test whether specific background manipulations (hypothesized types of availability) shift model feature reliance. As shown in Figure 6C, we find that Bird accuracy increases as we reduce the availability of the image background by manipulating its spatial extent (*Bird size*, *Background patch removal*) or drop background color (*Color*), implicating these as among the features that models latch onto in preferring image backgrounds (validated with explainability analyses in C.5). Experiments in Figure B.9 show that this phenomenon also occurs in ImageNet-pretrained models; background noise and spatial frequency manipulations also drive feature reliance.

## 8 Conclusion

Shortcut learning is of both scientific and practical interest given its implications for how models generalize. Why do some features become shortcut features? Here, we introduced the notion of

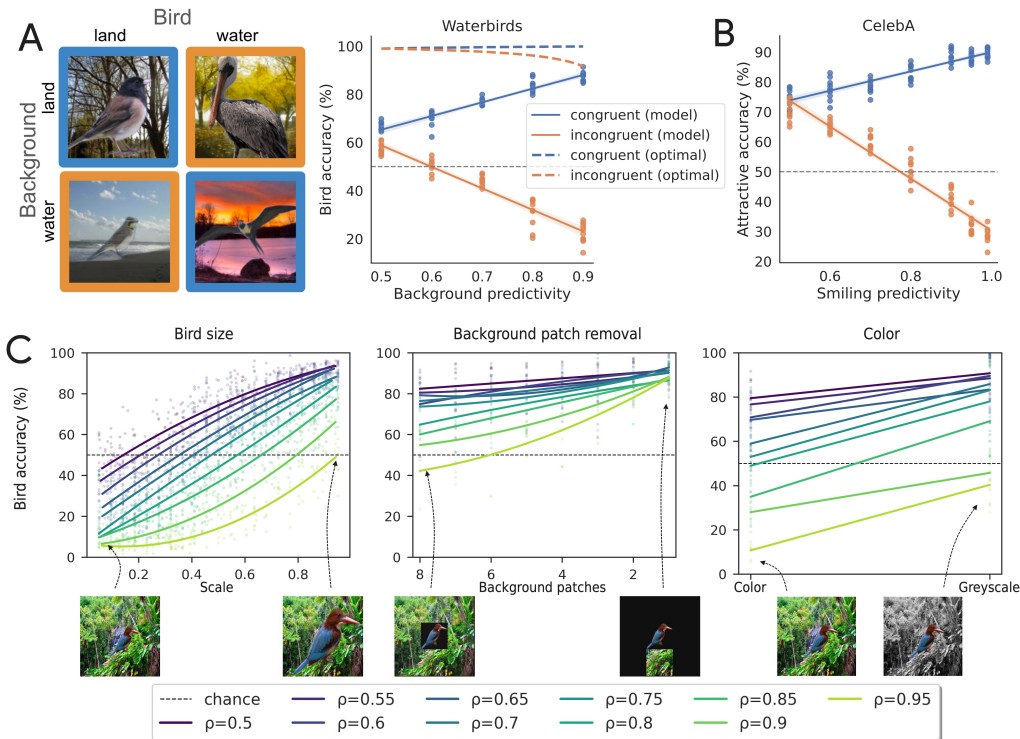

Figure 6: **Availability as well as predictivity determines which features image classifiers rely on.** A: Models (ResNet18) were trained to classify Birds from images sampled from Waterbirds. We varied Background (non-core) predictivity while keeping Bird (core) predictivity fixed (= 0.99), and show Bird classification accuracy for two types of probes: *congruent* (blue, core and non-core features support the same label) and *incongruent* (orange, core and non-core features support opposing labels). As Background predictivity increases, the gap in accuracy between incongruent and congruent probes also increases. The model is more sensitive to the non-core feature than expected by a Bayes-optimal classifier (*optimal*): predictivity alone does not explain the model's behavior. B: Similar to the Waterbirds dataset, models trained to classify images from CelebA as "Attractive" exhibit an effect of "Smiling" availability. C: Bird accuracy for incongruent Waterbirds probes is influenced by both Background predictivity ($\rho$) and availability when we manipulate the latter explicitly (see also B.9).

*availability* and conducted studies systematically varying availability. We proposed a generative framework that allows for independent manipulation of predictivity and availability of latent features. Testing hypotheses about the contributions of each to model behavior, we find that for both vector and image classification tasks, deep nonlinear models exhibit a shortcut bias, deviating from the statistically optimal classifier in their feature reliance. We provided a theoretical account which indicates the inevitability of a shortcut bias for a single-hidden-layer nonlinear (ReLU) MLP but not a linear one. The theory specifies the exact interaction between predictivity and availability, and consistent with our empirical studies, predicts that availability can trump predictivity. In naturalistic datasets, vision architectures used in practice rely on non-core features more than they should on statistical grounds alone. Connecting with prior work identifying availability biases for texture and image background, we explicitly manipulated background properties such as spatial extent, color, and spatial frequency and found that they influence a model's propensity to learn shortcuts.

Taken together, our empirical and theoretical findings highlight that models used in practice are prone to shortcut learning, and that to understand model behavior, one must consider the contributions of both feature predictivity and availability. Future work will study shortcut features in additional domains, and develop methods for automatically discovering further shortcut features which drive model behavior. The generative framework we have laid out will support a systematic investigation of architectural manipulations which may influence shortcut learning.

## ACKNOWLEDGMENTS

We thank Pieter-Jan Kindermans for feedback on the manuscript, and Jaehoon Lee, Lechao Xiao, Robert Geirhos, Olivia Wiles, Isabelle Guyon, and Roman Novak for interesting discussions.

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

# Supplementary Material for
# "On the Foundations of Shortcut Learning"

## A BROADER IMPACTS

Our work aims to achieve a principled and fundamental understanding of the mechanisms of deep learning—when it succeeds, when it fails, and why it fails. As the goal is to better understand existing algorithms, there is no downside risk of the research. The upside benefit concerns cases of shortcut learning which have been identified as socially problematic and where fairness issues are at play, for example, the use of shortcut features such as gender or ethnicity that might be spuriously correlated with a core feature (e.g., likelihood of default).

## B SUPPLEMENTARY FIGURES

Methods details accompanying these figures appear in Section C.

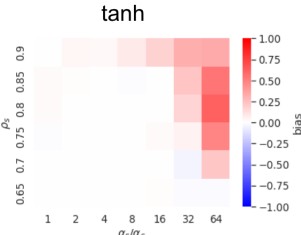

Figure B.1: Bias as a function of $z_s$ availability and predictivity for a single-hidden-layer MLP with a tanh hidden-layer activation function. Settings match those described in Figure 3C.

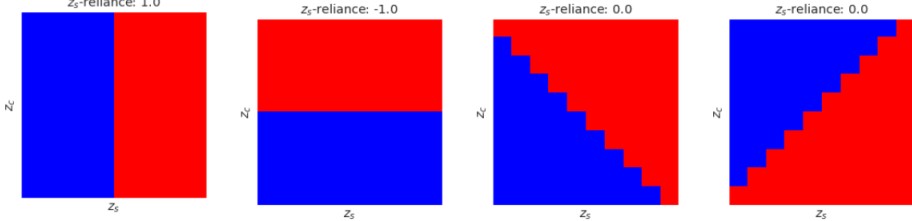

Figure B.2: The $z_s$-reliance (Sections 3 and C.1) of hypothetical classifiers which rely exclusively on $z_s$ (top left) or $z_c$ (top right), or rely equally on both features (bottom row). Color indicates the predicted label (red: pos, blue: neg) for probe items plotted by base probe $(z_s, z_c)$ values

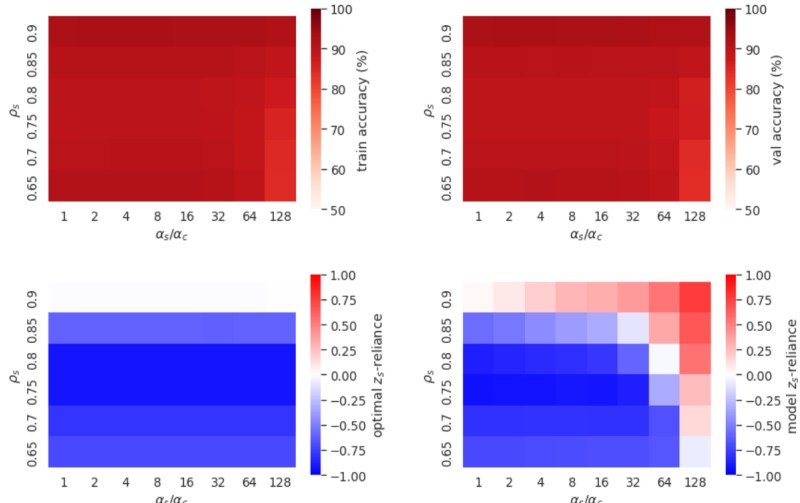

Figure B.3: Supplementary data for the results presented in Figure 2. TOP ROW: Model performance on the train (left) and validation (right) sets. BOTTOM ROW: $z_s$-reliance for the optimal classifier (left) and models (right).

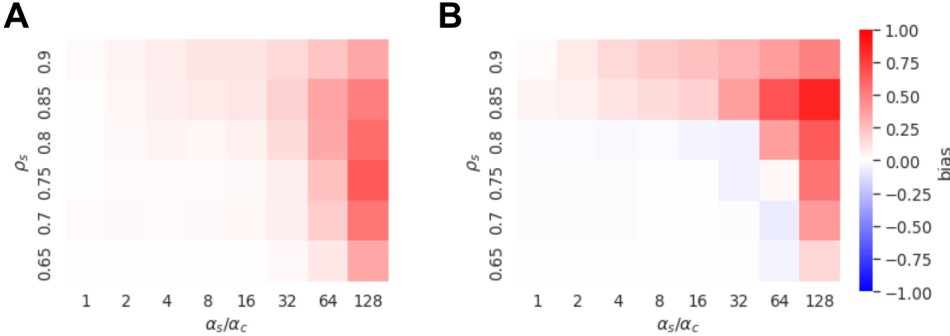

Figure B.4: The results shown in Figure 2 are not specific to choice of $\sigma_{sc}$. Results for the same experiment using datasets with $\sigma_{sc} = 0.4$ (A) and $\sigma_{sc} = 0.8$ (B).

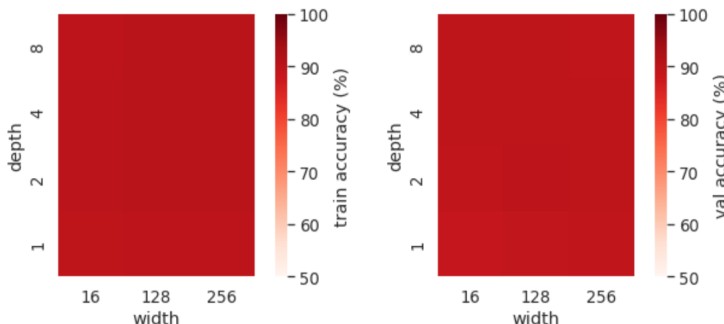

Figure B.5: Supplementary data for the results presented in Figure 3. Model performance on the train (left) and validation (right) sets.

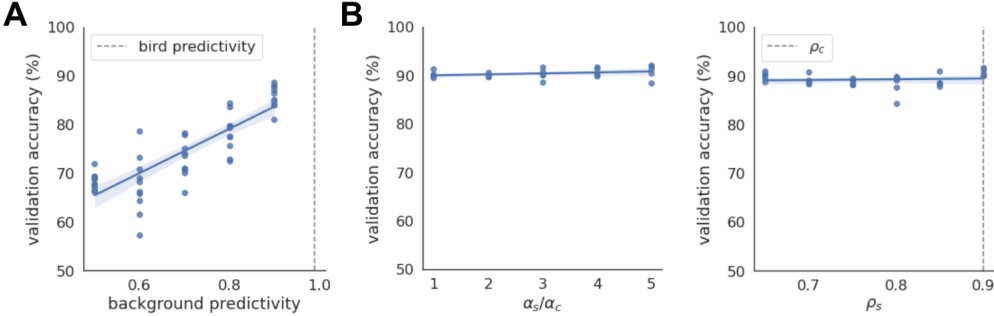

Figure B.6: Supplementary data for image experiments. Model performance underlying A: Experiments with the Waterbirds datasets presented in Figure 6 and described in C.5, and B: Experiments with image instantiations of the base dataset manipulating pixel footprint presented in Figure 4 B (left) and C (right).

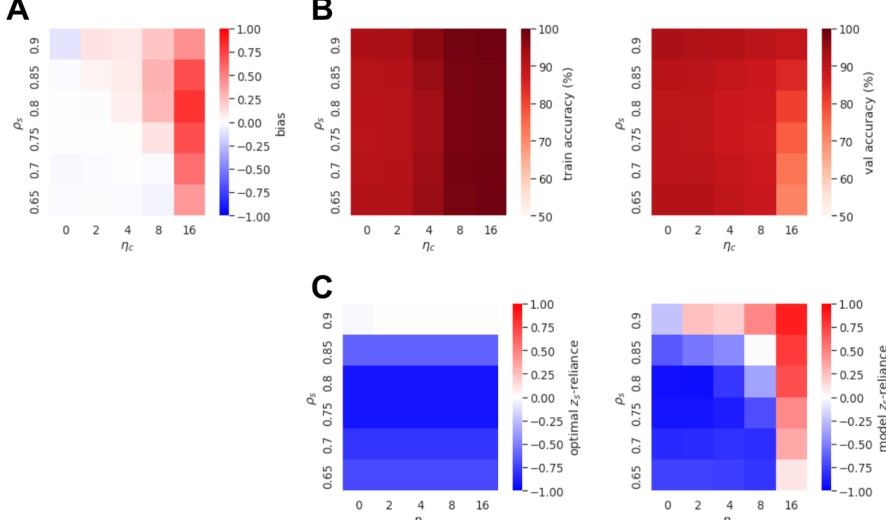

Figure B.7: **Models prefer a shallowly embedded nonlinear feature ($z_s$) to a deeply embedded nonlinear one** ($z_c$). A: The color of each cell of the heatmap indicates the mean bias of models as a function of the degree of nesting of $z_c$ ($\eta_c$), and the predictivity of $z_s$ ($\rho_s$). $z_s$ is always shallowly embedded ($\eta_s = 0$), and the predictivity of $z_c$ is held fixed ($\rho_c = 0.9$). See C.3 for additional details. B: Model performance. C:$z_s$-reliance of the optimal classifier (left) versus the model (right).

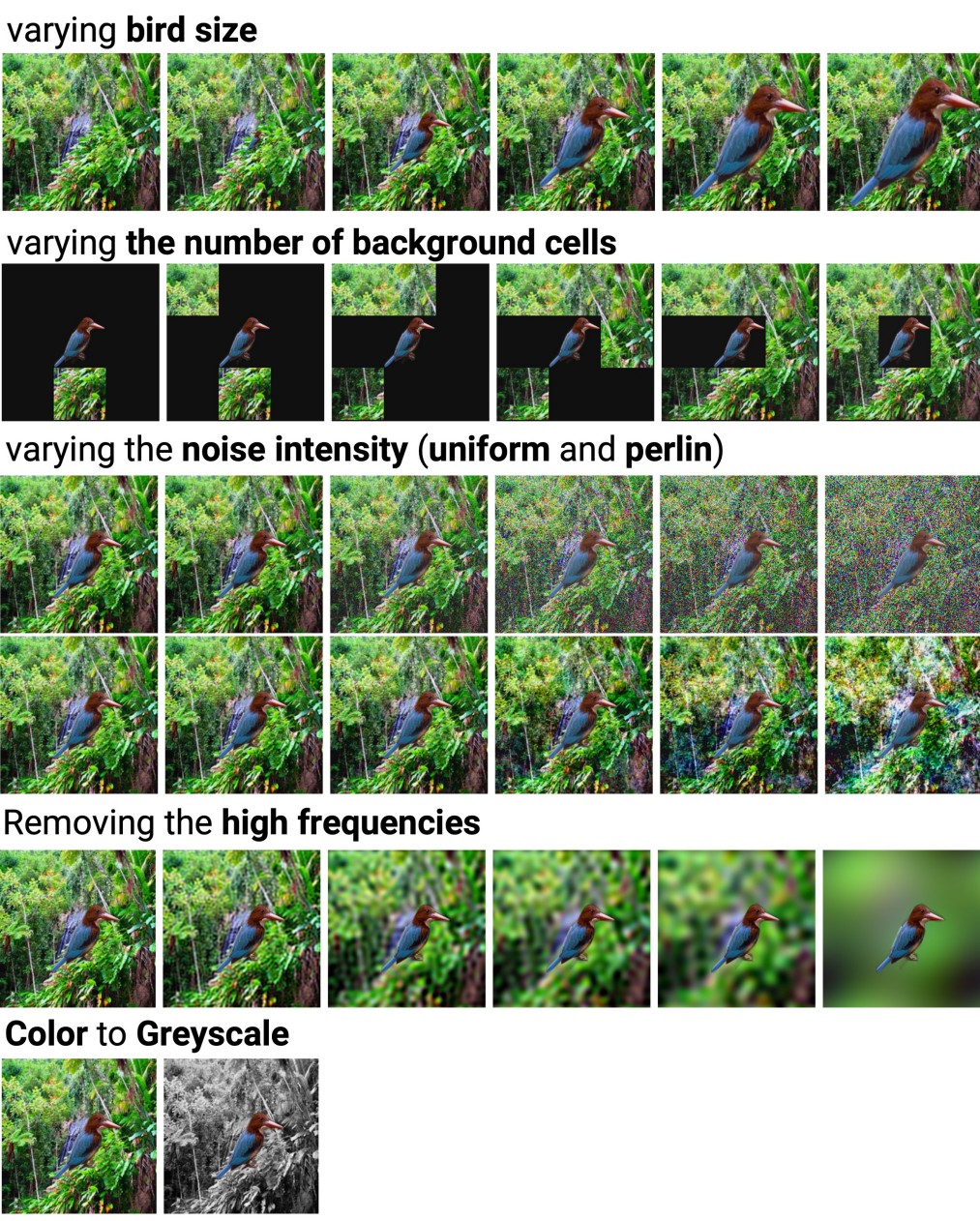

Figure B.8: **Illustration of studied Background availability manipulations.** For each of the six perturbations influencing the availability of both the background and the bird (in the case of bird scale), we provide visual representations of the potential perturbations applied to the training dataset. For instance, the first row demonstrates variations in the dataset resulting from changes in bird size, ranging from 5% to 95%.

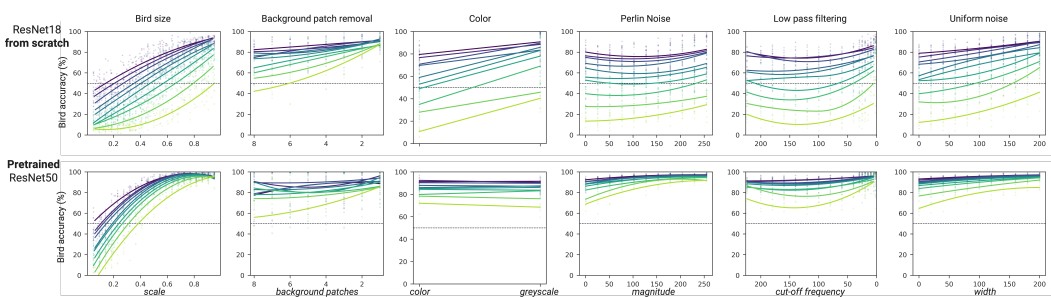

Figure B.9: **Manipulations of image backgrounds show that availability as well as predictivity determines feature reliance in vision models, including in models pretrained on ImageNet.** Expanding on the results shown in Figure 6, we plot Bird accuracy for incongruent probes (opposing Bird and Background labels) as a function of Background predictivity ($\rho$) and hypothesized types of availability. TOP ROW: ResNet18 models. BOTTOM ROW: IN ResNet50 models (see Section C.5 for additional details).

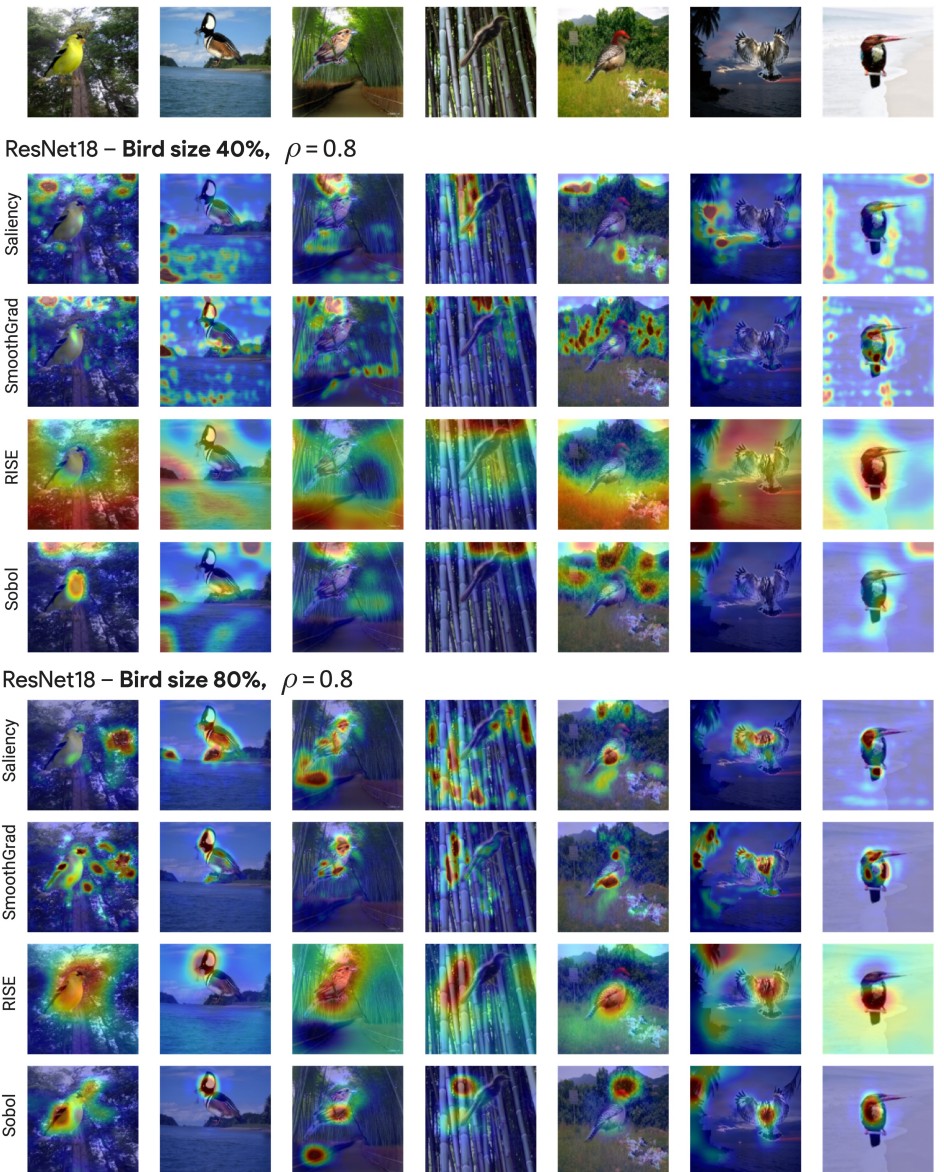

Figure B.10: **Explainability sanity check: Attribution maps corroborate Background availability study.** In Figures 6 and B.9, we saw that a Bird size manipulation affects models' tendency to classify incongruent probes consistent with their Bird labels. Here, we see that attribution maps for probe images (top row) reflect a difference in focal point for models trained on images with Bird size $= 40\%$ (preference for image background) versus Bird size $= 80\%$ (preference for foreground bird). See C.5 for additional details.

# C    SUPPLEMENTARY METHODS AND RESULTS

## C.1    ASSESSING BIAS

In Section 3, we described how we determine $reliance_\mathcal{M}$, a score which quantifies the extent to which a model (optimal classifier or trained model) uses feature $z_s$ as the basis for its classification decisions. In Figure B.2, for intuition, we show the $z_s$ reliance of hypothetical classifiers.

## C.2    SYNTHETIC DATA

**Optimal classifier.** To obtain optimal classifications, we use Linear Discriminant Analysis (LDA, as implemented in `Sklearn` with the least-squares solver). In all but the experiments on the effect of nesting as part of the generative process (C.3 and Figure B.7), the optimal classifier was fit to and probed with the same embedded base inputs used to train the corresponding model in a given experiment. In the nesting experiments, LDA was fit to and probed with base inputs directly.

## C.3    VECTOR EXPERIMENTS

**Default model architecture and training procedure.** Unless otherwise described, we train a multi-layer perceptron (MLP, depth = 8, width = 128, hidden activation function = ReLU) with MSE loss for 100 epochs using an SGD optimizer with batch size = 64 and learning rate = $1e − 02$. We use Glorot Normal weight initialization.

**Models prefer the more-available feature, even when it is less predictive.** In these experiments, given a base dataset with $(\rho_s, \rho_c)$, for each availability setting $(\alpha_s, \alpha_c)$, we manipulate the availability of the dataset, sample a random intialization seed, train the default model architecture, and evaluate the bias of the model. We repeat this process for each of 10 runs, computing the biases for individual (model, optimal classifier) pairs, and then averaging the results.

**Effect of nesting on availability**. Figure 1B illustrates how our generative procedure for synthetic datasets supports nested embedding of latent features, as described in Section 3. In this experiment, we test whether a model prefers a shallowly embedded nonlinear feature ($z_s$) to a deeply embedded nonlinear one ($z_c$) when the amplitude of both features is matched ($\alpha_s = \alpha_c = 0.1$) and the nonlinearity is a scaled $\tanh$ ($\tanh(\lambda x)$, with $\lambda = 100$). After embedding each feature as $e_i = \alpha_i w_i z_i$, we apply the nonlinearity. For a nesting factor $\eta_i > 0$, we pass $e_i$ through a fully connected network (depth = $\eta_i$, width = $d$, no bias weights) with scaled $\tanh$ activations on the hidden and output layers. We initialize weight matrices to be random special orthogonal matrices (implemented as `scipy.stats.special_ortho_group`). In Figure B.7, we show bias as a function of $\eta_c$ and $\rho_s$; $z_s$ is fixed to be shallowly embedded with $\eta_s = 0$ and has $\rho_c = 0.9$. We report the results as the mean across 3 runs.

## C.4    PIXEL FOOTPRINT EXPERIMENTS

**Model architecture and training procedure.** In these experiments, we train a randomly initialized ResNet18 architecture with MSE loss using an Adam optimizer with batch size = 64 and learning rate = $1e − 02$, taking the best (defined on validation accuracy) across 100 epochs of training. For each $(\rho_s, \rho_c)$ predictivity setting, we repeat this process for 10 (sampled dataset, random weight initialization) runs.

**Analyses.** In Figure 4, we report results for experiments in which we use a base $z_s$ pixel footprint of 400 px, and manipulate $z_s$ pixel footprint to be $1 − 5\times$ larger ($\alpha_c = 1$, $\alpha_s \in [1, 5]$). To determine the bias of a model, we compute the $z_s$-reliance of the trained model given image instantiations of the base probe items, and compare to the $z_s$-reliance of the optimal classifier (LDA) fit to and probed using vector instantiations of the same base dataset with the same $\alpha_s$ and $\alpha_c$. We repeat this process for 5 runs, computing bias on a per-run basis as in Section C.3.

## C.5    FEATURE AVAILABILITY IN NATURALISTIC DATASETS

**Datasets.** *Waterbirds*. Waterbirds images (Sagawa et al., 2020a) combine birds taken from the Caltech-UCSD Birds-200-2011 dataset (Wah et al., 2011) and backgrounds from the Places dataset

(Zhou et al., 2017). To construct the datasets used in Figure 6A, we sample images from a base Waterbirds dataset generated using code by (Sagawa et al., 2020a) (`github.com/kohpangwei/group_DRO`) with `val frac` = 0.2 and `confounder strength` = 0.5, yielding sets of 5694 train images ($224 \times 224$), 300 validation images, and 5794 test images. We then subsample these sets to 1200 train, 90 validation ($\times 2$ sets), and 1000 probe images ($\times 2$ sets), respectively, such that the train and validation sets instantiate target feature predictivities, and the probe sets contain congruent or incongruent feature-values, as described in Section 7.

To construct the datasets used in Figures 6C and B.9, for a given predictivity setting, we subsample 8 bird types per category (land, water) and cross them with land and water backgrounds randomly sampled from the standard Waterbirds background classes, to generate class-balanced train sets ranging in size from $800 - 1200$, and incongruent probe sets of size $400$.

*CelebA*. The CelebA (Liu et al., 2015) dataset contains images of celebrity faces paired with a variety of binary attribute labels. In our experiments, we cast the "Attractive" label as the core feature, and the "Smiling" label as the non-core feature. To construct the datasets used in Figure 6B, we sample images consistent with the target predictivities.

**Waterbirds availability manipulations**. In the datasets depicted in Figures 6C and B.9, we aim to manipulate various aspects of background availability. To do so, we use CUB-200 masks to exclusively modify the background, and apply five distinct types of perturbations: altering bird size, removing background patches, changing background colors, applying low-pass filtering, adding Perlin noise, and adding white noise. These perturbations test hypotheses about specific image properties than may make an image background more available to a model than the foreground object. We vary Background predicitivity while holding bird predictivity fixed ($= 0.99$). We report results for at least 500 runs (model training) for each perturbation type. Figure B.8 shows sample image to which the perturbations have been applied. In detail, the perturbations are as follow:

- **Bird size**: We manipulate the pixel footprint of the bird by scaling the square mask. Note that an increase in bird size corresponds to a decrease in background pixels in the image.

- **Background patch removal**: This manipulation removes patches of the image background while preserving the foreground bird. To apply the manipulation, we partition the image into a $3 \times 3$ checkerboard, position the bird in the center, and then remove $x$ in $[1, 8]$ background cells.

- **Color**: To assess the influence of color information, we use the original image with its colored background ("Color") or convert the background to grayscale ("Grayscale").

- **Low-pass filtering**: We apply a low-pass filter to the frequency representation of the background, with the cutoff frequency varying from 224 (original image) to 2 (retaining only components at 2 cycles per image).

- **Uniform noise**: Uniform noise is introduced to the image with an amplitude ranging from 0 (original image) to 255. It is essential to note that image values are clipped within the (0, 255) range before preprocessing.

- **Perlin noise**: We apply Perlin noise, which possesses spatial coherence, at various intensity levels along with uniform noise. We employ 6-octave Perlin noise to match the frequency distribution of natural backgrounds (wavelengths 2, 4, 8, 16, 32, 64, and 128).

**Model architecture and training procedure.** For the experiments shown in Figure 6A and B, we normalize all images by the ImageNet train-set statistics (RGB mean $= [0.485, 0.456, 0.406]$, std $= [0.229, 0.224, 0.225]$), and then use the same settings as described in 5.

In the experiments shown in Figures 6C and B.9, we preprocess images by normalizing to be in $[-1, 1]$, apply random crops of size 200 | 200, resize to 224 | 224, and randomly flip over the horizontal axis with $p = 0.5$ We train models with MSE loss using an Adam optimizer with batch size $= 32$, cosine decay learning rate schedule (linear warmup, initial learning rate $= 1e - 03$), and weight decay $= 1e - 05$. For the ResNet18 experiments in Figures 6C and B.9, we train a randomly initialized ResNet18 trained for 30 epochs. For the ResNet50 experiments in Figure B.9, we train the randomly initialized readout layer of a frozen, ImageNet-pretrained ResNet50 (*IN ResNet50*) for 15 epochs.

**Results: Vision models, including ImageNet-pretrained ones, are sensitive to Background availability manipulations.** Figure B.9 displays the results of the six manipulations which we hypothesized affect Background availability. The top row shows accuracy for ResNet18 models trained from scratch, while the bottom row shows the results for IN-ResNet50. Our findings reveal several key observations:

First, as the size of the bird increases (Col.1), as well as when the background becomes noisier (Col. 4,6), filtered of its high frequencies (Col 5), or simply masked (Col 2), the model tends to rely more heavily on the bird itself for classification. Conversely, when the background is clear and informative (available), the model utilizes the background features to predict the class. Second, the spuriousness of the background, as quantified by $\rho$, also has a discernible effect on the model's behavior. However, it only partially accounts for the observed variations, demonstrating that availability factors are at play. Last, these results indicate that pretraining has a beneficial impact and can modulate feature availability. This effect may be attributed to the learned invariances during the pretraining process, enabling the model to better adapt to different availability conditions. Our spatial frequency experiments complement existing and concurrent work studying what models learn as a function of spatial frequency manipulations at training (Jo & Bengio, 2017; Yin et al., 2019; Tsuzuku & Sato, 2019; Hermann et al., 2020; Subramanian et al., 2023) or test time (Geirhos et al., 2018b).

**Results: Explainability analyses corroborate Background availability study.** The experiments presented in Figures 6C and B.9 showed that the degree of Background availability influences model classifications of incongruent probes. Here, we use attribution methods to verify that when a model predominantly relies on the image background, or on the foreground bird, according to the experimental results, its focal point reflect this. Figure B.10 shows the attributions maps of two ResNet18 models trained from scratch, one using a bird size of 40%, and the other using a bird size of 80%. For the same set of probe images, the model trained on larger birds exhibits a strategy shift in classification, seemingly having his focal point on the birds, whereas the model trained on smaller birds mainly relies on the background. The methods employed encompass a combination of black-box techniques (Rise (Petsiuk et al., 2018) and Sobol indices (Fel et al., 2021)) and gradient-based approaches (Saliency (Zeiler & Fergus, 2014) and Smoothgrad (Smilkov et al., 2017)).

# D  QUADRATIC APPROXIMATION OF $h$

Since $u$ is the dot product of two normalized vectors, its range is limited to $-1 \leq u \leq 1$. Observe that the derivative of $h$ has the form,

$$h'(u) = 1 - \frac{1}{\pi} \arccos(u).$$ (8)

The only place within $u \in [-1, 1]$ that $h'$ vanishes is at $u = -1$, suggesting the critical point of our quadratic must lie at this point. Furthermore, observe that $h(-1) = 0$. Forcing these two constraints onto our quadratic, leaves us with the form,

$$\widehat{h}(u) = a(1 + u)^2,$$ (9)

where $a$ is a parameter to be determined. Our aim is to find $a$ so that the function $\widehat{h}$ and $h$ looks similar over the entire domain $u \in [-1, 1]$. We choose $\ell_2$ error between the two functions defined as,

$$e_a = \int_{-1}^{1} \left( h(u) - \widehat{h}(u) \right)^2 d\,u.$$ (10)

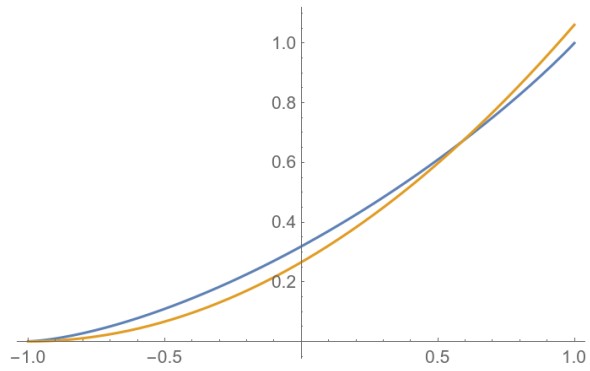

Figure D.1: The plots for $h$ (blue) $\widehat{h}$ (orange) within $u \in [-1, 1]$.

Replacing the definitions for $h$ and $\widehat{h}$ into $(h(u) - \widehat{h}(u))^2$, one can [1] obtain

$$e_a = \frac{1}{3} - \frac{163}{48}a + \frac{32}{5}a^2 + \frac{32}{27\pi^2}. \tag{18}$$

Since $e_a$ is a convex quadratic in $a$, its minimizer can be determined by zero-crossing the derivative of $e_a$ w.r.t. $a$. This yields,

$$a^* \triangleq \arg\min_a e_a = \frac{815}{3072}. \tag{19}$$

This leads to the claimed quadratic form,

$$\widehat{h}(u) \triangleq \frac{815}{3072}(1 + u)^2. \tag{20}$$

Figure D.1 shows a plot of the original $h$ and the quadratic approximation $\widehat{h}$ to visually get a sense of the approximation quality.

## E    PROOF OF ReLU KERNEL THEOREM

Recall that the kernel function $k$ for ReLU is expressed using $h$ as in (3). Replacing $h$ with $\widehat{h}$ thus gets us an alternate kernel function which approximates that of ReLU's.

$$k(\boldsymbol{x}, \boldsymbol{z}) \triangleq \|\boldsymbol{x}\| \|\boldsymbol{z}\| \widehat{h}\left(\left\langle \frac{\boldsymbol{x}}{\|\boldsymbol{x}\|}, \frac{\boldsymbol{z}}{\|\boldsymbol{z}\|} \right\rangle\right) = \|\boldsymbol{x}\| \|\boldsymbol{z}\| a^*\left(1 + \left(\left\langle \frac{\boldsymbol{x}}{\|\boldsymbol{x}\|}, \frac{\boldsymbol{z}}{\|\boldsymbol{z}\|} \right\rangle\right)\right)^2. \tag{21}$$

---

[1]We first claim that the indefinite integral of $(h(u) - \widehat{h}(u))^2$ has the form,

$$E_a(u) \triangleq \int \left(\frac{1}{\pi}\left(u\left(\pi - \arccos(u)\right) + \sqrt{1 - u^2}\right) - a(1 + u)^2\right)^2 du \tag{11}$$

$$= \frac{1}{2160\pi^2}\left(432\pi^2 a^2 u^5 + 80\left(9\pi^2\left(6a^2 - 4a + 1\right) - 17\right)u^3\right. \tag{12}$$

$$+ 240\left(9\pi^2 a^2 + 11\right)u + 1080\pi^2 a(2a - 1)u^4 + 2160\pi^2 a(2a - 1)u^2 \tag{13}$$

$$- 15\pi\sqrt{1 - u^2}\left(a\left(90u^3 + 256u^2 + 207u - 64\right) - 128u^2 + 32\right) \tag{14}$$

$$+ 120\left(3\pi a\left(3u^2 + 8u + 6\right)u^2 - 12\pi u^3 + 4\left(1 - 4u^2\right)\sqrt{1 - u^2}\right)\cos^{-1}(u) \tag{15}$$

$$\left. - 1215\pi a \sin^{-1}(u) + 720u^3 \cos^{-1}(u)^2\right) \tag{16}$$

This can be be easily shown by taking the derivative of both sides and arriving at equality. The definite integral can now computed by evaluation the indefinite integral at the boundary points.

$$e_a = E_a(1) - E_a(-1) = \frac{1}{3} - \frac{163}{48}a + \frac{32}{5}a^2 + \frac{32}{27\pi^2}. \tag{17}$$

### E.1 EIGENFUNCTIONS

An eigenfunction $\phi$ must satisfy,

$$\int_{\mathbb{R}^n} a^* \|\boldsymbol{x}\| \|\boldsymbol{z}\| \left(1 + \left\langle \frac{\boldsymbol{x}}{\|\boldsymbol{x}\|}, \frac{\boldsymbol{z}}{\|\boldsymbol{z}\|} \right\rangle\right)^2 \phi(\boldsymbol{z})p(\boldsymbol{z}) \, d\boldsymbol{z} \quad = \quad \lambda \, \phi(\boldsymbol{x}), \tag{22}$$

or equivalently,

$$a^* \|\boldsymbol{x}\| \int_{\mathbb{R}^n} \|\boldsymbol{z}\| \left(1 + \left\langle \frac{\boldsymbol{x}}{\|\boldsymbol{x}\|}, \frac{\boldsymbol{z}}{\|\boldsymbol{z}\|} \right\rangle\right)^2 \phi(\boldsymbol{z})p(\boldsymbol{z}) \, d\boldsymbol{z} \quad = \quad \lambda \, \phi(\boldsymbol{x}), \tag{23}$$

We first focus on the above integral.

$$\int_{\mathbb{R}^n} \|\boldsymbol{z}\| \left(1 + \left\langle \frac{\boldsymbol{x}}{\|\boldsymbol{x}\|}, \frac{\boldsymbol{z}}{\|\boldsymbol{z}\|} \right\rangle\right)^2 \phi(\boldsymbol{z})p(\boldsymbol{z}) \, d\boldsymbol{z} \tag{24}$$

$$= \int_{\mathbb{R}^n} \|\boldsymbol{z}\| \left(1 + 2\frac{\langle \boldsymbol{x}, \boldsymbol{z} \rangle}{\|\boldsymbol{x}\| \|\boldsymbol{z}\|} + \frac{\langle \boldsymbol{x}, \boldsymbol{z} \rangle^2}{\|\boldsymbol{x}\|^2 \|\boldsymbol{z}\|^2}\right) \phi(\boldsymbol{z})p(\boldsymbol{z}) \, d\boldsymbol{z} \tag{25}$$

$$= \int_{\mathbb{R}^n} \left(\|\boldsymbol{z}\| + 2\frac{\langle \boldsymbol{x}, \boldsymbol{z} \rangle}{\|\boldsymbol{x}\|} + \frac{\langle \boldsymbol{x}, \boldsymbol{z} \rangle^2}{\|\boldsymbol{x}\|^2 \|\boldsymbol{z}\|}\right) \phi(\boldsymbol{z})p(\boldsymbol{z}) \, d\boldsymbol{z} \tag{26}$$

$$= \int_{\mathbb{R}^2} \left(\|\boldsymbol{z}\| + 2\frac{x_1 z_1 + x_2 z_2}{\sqrt{x_1^2 + x_2^2}} + \frac{x_1^2 z_1^2 + x_2^2 z_2^2 + 2x_1 x_2 z_1 z_2}{(x_1^2 + x_2^2)\sqrt{z_1^2 + z_2^2}}\right) \phi(\boldsymbol{z})p(\boldsymbol{z}) \, d\boldsymbol{z} \tag{27}$$

$$= \int_{\mathbb{R}^2} \left(\|\boldsymbol{z}\| + \sqrt{2}z_1 \frac{\sqrt{2}x_1}{\sqrt{x_1^2 + x_2^2}} + \sqrt{2}z_2 \frac{\sqrt{2}x_2}{\sqrt{x_1^2 + x_2^2}} + \frac{z_1^2}{\sqrt{z_1^2 + z_2^2}} \frac{x_1^2}{(x_1^2 + x_2^2)}\right. \tag{28}$$

$$\left. + \frac{z_2^2}{\sqrt{z_1^2 + z_2^2}} \frac{x_2^2}{(x_1^2 + x_2^2)} + \frac{\sqrt{2}z_1 z_2}{\sqrt{z_1^2 + z_2^2}} \frac{\sqrt{2}x_1 x_2}{(x_1^2 + x_2^2)}\right) \phi(\boldsymbol{z})p(\boldsymbol{z}) \, d\boldsymbol{z} \tag{29}$$

$$= a_0 + a_1 \frac{\sqrt{2}x_1}{\sqrt{x_1^2 + x_2^2}} + a_2 \frac{\sqrt{2}x_2}{\sqrt{x_1^2 + x_2^2}} + a_3 \frac{x_1^2}{x_1^2 + x_2^2} + a_4 \frac{x_2^2}{x_1^2 + x_2^2} + a_5 \frac{\sqrt{2}x_1 x_2}{x_1^2 + x_2^2}, \tag{30}$$

where $a_0$ to $a_5$ are the result of the definite integral for each term. Observe that each $a_i$ is thus constant in $\boldsymbol{x}$ and $\boldsymbol{z}$. Plugging the above expression into (23) yields,

$$a^* \|\boldsymbol{x}\| (a_0 + a_1 \frac{\sqrt{2}x_1}{\sqrt{x_1^2 + x_2^2}} + a_2 \frac{\sqrt{2}x_2}{\sqrt{x_1^2 + x_2^2}} + a_3 \frac{x_1^2}{x_1^2 + x_2^2} + a_4 \frac{x_2^2}{x_1^2 + x_2^2} + a_5 \frac{\sqrt{2}x_1 x_2}{x_1^2 + x_2^2}) \quad = \quad \lambda \, \phi(\boldsymbol{x}), \tag{31}$$

Thus the eigenfunction has the form,

$$\phi(\boldsymbol{x}) = \frac{a^*}{\lambda} \|\boldsymbol{x}\| (a_0 + a_1 \frac{\sqrt{2}x_1}{\sqrt{x_1^2 + x_2^2}} + a_2 \frac{\sqrt{2}x_2}{\sqrt{x_1^2 + x_2^2}} + a_3 \frac{x_1^2}{x_1^2 + x_2^2} + a_4 \frac{x_2^2}{x_1^2 + x_2^2} + a_5 \frac{\sqrt{2}x_1 x_2}{x_1^2 + x_2^2}). \tag{32}$$

In order to figure out the values of $a_0$ to $a_5$ we plug the obtained eigenfunction form into (23) again, for both $\phi(\boldsymbol{x})$ and $\phi(\boldsymbol{z})$, That takes us from

$$a^* \|\boldsymbol{x}\| \int_{\mathbb{R}^2} \|\boldsymbol{z}\| \left(1 + \left\langle \frac{\boldsymbol{x}}{\|\boldsymbol{x}\|}, \frac{\boldsymbol{z}}{\|\boldsymbol{z}\|} \right\rangle\right)^2 \phi(\boldsymbol{z})p(\boldsymbol{z}) \, d\boldsymbol{z} \quad = \quad \lambda \, \phi(\boldsymbol{x}), \tag{33}$$

to,

$$a^* \|\boldsymbol{x}\| \int_{\mathbb{R}^2} \left(\|\boldsymbol{z}\| + \sqrt{2}z_1 \frac{\sqrt{2}x_1}{\sqrt{x_1^2 + x_2^2}} + \sqrt{2}z_2 \frac{\sqrt{2}x_2}{\sqrt{x_1^2 + x_2^2}} + \frac{z_1^2}{\sqrt{z_1^2 + z_2^2}} \frac{x_1^2}{(x_1^2 + x_2^2)}\right. \tag{34}$$

$$\left. + \frac{z_2^2}{\sqrt{z_1^2 + z_2^2}} \frac{x_2^2}{(x_1^2 + x_2^2)} + \frac{\sqrt{2}z_1 z_2}{\sqrt{z_1^2 + z_2^2}} \frac{\sqrt{2}x_1 x_2}{(x_1^2 + x_2^2)}\right) \tag{35}$$

$$\times \frac{a^*}{\lambda} \|\boldsymbol{z}\| (a_0 + a_1 \frac{\sqrt{2}z_1}{\sqrt{z_1^2 + z_2^2}} + a_2 \frac{\sqrt{2}z_2}{\sqrt{z_1^2 + z_2^2}} + a_3 \frac{z_1^2}{z_1^2 + z_2^2} + a_4 \frac{z_2^2}{z_1^2 + z_2^2} + a_5 \frac{\sqrt{2}z_1 z_2}{z_1^2 + z_2^2})$$

$$\times p(\boldsymbol{z}) \, d\boldsymbol{z} \tag{36}$$

$$= \lambda \frac{a^*}{\lambda} \|\boldsymbol{x}\| (a_0 + a_1 \frac{\sqrt{2}x_1}{\sqrt{x_1^2 + x_2^2}} + a_2 \frac{\sqrt{2}x_2}{\sqrt{x_1^2 + x_2^2}} + a_3 \frac{x_1^2}{x_1^2 + x_2^2} + a_4 \frac{x_2^2}{x_1^2 + x_2^2} + a_5 \frac{\sqrt{2}x_1 x_2}{x_1^2 + x_2^2}). \tag{37}$$

Dividing both sides by $a^* \|\boldsymbol{x}\|/\lambda$ yields,

$$a^* \|\boldsymbol{x}\| \int_{\mathbb{R}^2} \|\boldsymbol{z}\| \left(1 + \left\langle \frac{\boldsymbol{x}}{\|\boldsymbol{x}\|}, \frac{\boldsymbol{z}}{\|\boldsymbol{z}\|} \right\rangle \right)^2 \phi(\boldsymbol{z}) p(\boldsymbol{z}) \, d\boldsymbol{z} \;=\; \lambda \, \phi(\boldsymbol{x}) \tag{38}$$

$$\Rightarrow \quad a^* \int_{\mathbb{R}^2} \left( \|\boldsymbol{z}\| + \sqrt{2} z_1 \frac{\sqrt{2} x_1}{\sqrt{x_1^2 + x_2^2}} + \sqrt{2} z_2 \frac{\sqrt{2} x_2}{\sqrt{x_1^2 + x_2^2}} + \frac{z_1^2}{\sqrt{z_1^2 + z_2^2}} \frac{x_1^2}{(x_1^2 + x_2^2)} \right. \tag{39}$$

$$\left. + \frac{z_2^2}{\sqrt{z_1^2 + z_2^2}} \frac{x_2^2}{(x_1^2 + x_2^2)} + \frac{\sqrt{2} z_1 z_2}{\sqrt{z_1^2 + z_2^2}} \frac{\sqrt{2} x_1 x_2}{(x_1^2 + x_2^2)} \right) \tag{40}$$

$$\times \|\boldsymbol{z}\| \, (a_0 + a_1 \frac{\sqrt{2} z_1}{\sqrt{z_1^2 + z_2^2}} + a_2 \frac{\sqrt{2} z_2}{\sqrt{z_1^2 + z_2^2}} + a_3 \frac{z_1^2}{z_1^2 + z_2^2} + a_4 \frac{z_2^2}{z_1^2 + z_2^2} + a_5 \frac{\sqrt{2} z_1 z_2}{z_1^2 + z_2^2}) \; p(\boldsymbol{z}) \, d\boldsymbol{z} \tag{41}$$

$$= \lambda \, (a_0 + a_1 \frac{\sqrt{2} x_1}{\sqrt{x_1^2 + x_2^2}} + a_2 \frac{\sqrt{2} x_2}{\sqrt{x_1^2 + x_2^2}} + a_3 \frac{x_1^2}{x_1^2 + x_2^2} + a_4 \frac{x_2^2}{x_1^2 + x_2^2} + a_5 \frac{\sqrt{2} x_1 x_2}{x_1^2 + x_2^2}) \,. \tag{42}$$

For the above identity to hold at any point $\boldsymbol{x}$ we need to equate the coefficients for each term involving $\boldsymbol{x}$. That is, we need to have,

$$a^* \left( \int_{\mathbb{R}^2} \left( \begin{bmatrix} \|\boldsymbol{z}\| \\ \sqrt{2} z_1 \\ \sqrt{2} z_2 \\ \frac{z_1^2}{\sqrt{z_1^2 + z_2^2}} \\ \frac{z_2^2}{\sqrt{z_1^2 + z_2^2}} \\ \frac{\sqrt{2} z_1 z_2}{\sqrt{z_1^2 + z_2^2}} \end{bmatrix} \begin{bmatrix} \|\boldsymbol{z}\| & \sqrt{2} z_1 & \sqrt{2} z_2 & \frac{z_1^2}{\sqrt{z_1^2 + z_2^2}} & \frac{z_2^2}{\sqrt{z_1^2 + z_2^2}} & \frac{\sqrt{2} z_1 z_2}{\sqrt{z_1^2 + z_2^2}} \end{bmatrix} \right) p(\boldsymbol{z}) \, d\boldsymbol{z} \right) \begin{bmatrix} a_0 \\ a_1 \\ a_2 \\ a_3 \\ a_4 \\ a_5 \end{bmatrix} = \lambda \begin{bmatrix} a_0 \\ a_1 \\ a_2 \\ a_3 \\ a_4 \\ a_5 \end{bmatrix} \tag{43}$$

or in a more compact form,

$$a^* \left( \int_{\mathbb{R}^2} \left( \boldsymbol{v}_{\boldsymbol{z}} \boldsymbol{v}_{\boldsymbol{z}}^T \right) p(\boldsymbol{z}) \, d\boldsymbol{z} \right) \boldsymbol{a} \;=\; \lambda \, \boldsymbol{a} \,, \tag{44}$$

where,

$$\boldsymbol{v}_{\boldsymbol{z}} \;\triangleq\; \begin{bmatrix} \|\boldsymbol{z}\| & \sqrt{2} z_1 & \sqrt{2} z_2 & \frac{z_1^2}{\|\boldsymbol{z}\|} & \frac{z_2^2}{\|\boldsymbol{z}\|} & \frac{\sqrt{2} z_1 z_2}{\|\boldsymbol{z}\|} \end{bmatrix}^T \,. \tag{45}$$

We now focus on the form of $p(\boldsymbol{z})$. Recall that we assume $p(\boldsymbol{z})$ is a mixture of $I$ Gaussian sources. Denote the selection probability of each Gaussian component by $\pi_i$ and the corresponding component-conditional normal density function by $g(\boldsymbol{z}\,;\,\boldsymbol{\mu}^{(i)}, \boldsymbol{C}^{(i)})$. Thus,

$$p(\boldsymbol{z}) \;=\; \sum_i \pi_i g(\boldsymbol{z}\,;\,\boldsymbol{\mu}^{(i)}, \boldsymbol{C}^{(i)}) \,. \tag{46}$$

Since we assume that the covariance matrix of each $g_i$ is small element-wise, we can resort to a Laplace-type approximation for the integrals as below.

$$\int_{\mathbb{R}^2} f(\boldsymbol{z}) \, p(\boldsymbol{z}) \, d\boldsymbol{z} \approx \sum_i \pi_i \, f(\boldsymbol{\mu}^{(i)}) \,. \tag{47}$$

Under this approximation, the linear system can be expressed as,

$$\underbrace{\left( \sum_i \pi_i \boldsymbol{v}_i \boldsymbol{v}_i^T \right)}_{\boldsymbol{C}} \boldsymbol{a} \;=\; \frac{\lambda}{a^*} \, \boldsymbol{a} \,, \tag{48}$$

where,

$$\boldsymbol{v}_i \;\triangleq\; \begin{bmatrix} \|\boldsymbol{\mu}^{(i)}\| & \sqrt{2} \mu_1^{(i)} & \sqrt{2} \mu_2^{(i)} & \frac{\mu_1^{(i)2}}{\|\boldsymbol{\mu}^{(i)}\|} & \frac{\mu_2^{(i)2}}{\|\boldsymbol{\mu}^{(i)}\|} & \frac{\sqrt{2} \mu_1^{(i)} \mu_2^{(i)}}{\|\boldsymbol{\mu}^{(i)}\|} \end{bmatrix}^T \,. \tag{49}$$

Thus, any solution $\boldsymbol{a}$ must be an eigenvector of $\boldsymbol{C}$. In the sequel we discuss how to obtain the eigenvectors of $\boldsymbol{C}$. In particular, in our 2-d problem with 2 Gaussian mixtures, where $\pi_1 = \pi_2 = 1/2$ and $\boldsymbol{\mu}^{(1)} = \boldsymbol{A}\boldsymbol{\mu}$ and $\boldsymbol{\mu}^{(2)} = -\boldsymbol{A}\boldsymbol{\mu}$, the linear equation can be expressed as below.

$$\underbrace{\left( \frac{1}{2} \boldsymbol{v}^+ \boldsymbol{v}^{+T} + \frac{1}{2} \boldsymbol{v}^- \boldsymbol{v}^{-T} \right)}_{\boldsymbol{C}} \boldsymbol{a} \;=\; \frac{\lambda}{a^*} \boldsymbol{a} \,, \tag{50}$$

where,

$$\boldsymbol{v}^+ \triangleq \left[ \|\boldsymbol{A\mu}\| \quad \sqrt{2}a_1\mu_1 \quad \sqrt{2}a_2\mu_2 \quad \frac{(a_1\mu_1)^2}{\|\boldsymbol{A\mu}\|} \quad \frac{(a_2\mu_2)^2}{\|\boldsymbol{A\mu}\|} \quad \frac{\sqrt{2}a_1\mu_1 a_2\mu_2}{\|\boldsymbol{A\mu}\|} \right]^T \tag{51}$$

$$\boldsymbol{v}^- \triangleq \left[ \|\boldsymbol{A\mu}\| \quad -\sqrt{2}a_1\mu_1 \quad -\sqrt{2}a_2\mu_2 \quad \frac{(a_1\mu_1)^2}{\|\boldsymbol{A\mu}\|} \quad \frac{(a_2\mu_2)^2}{\|\boldsymbol{A\mu}\|} \quad \frac{\sqrt{2}a_1\mu_1 a_2\mu_2}{\|\boldsymbol{A\mu}\|} \right]^T . \tag{52}$$

We now seek the eigenvectors of $\boldsymbol{C}$ in this specific setting. A key observation is that $\|\boldsymbol{v}^+\| = \|\boldsymbol{v}^-\|$ because the two are different only in the sign of their components. We now claim that, for any two vectors $\boldsymbol{v}^+$ and $\boldsymbol{v}^-$ such that $\|\boldsymbol{v}^+\| = \|\boldsymbol{v}^-\|$. the eigevectors of the matrix $\boldsymbol{C} \triangleq \boldsymbol{v}^+\boldsymbol{v}^{+^T} + \boldsymbol{v}^-\boldsymbol{v}^{-^T}$ have the form,

$$\boldsymbol{\psi}_1 = c_1 (\boldsymbol{v}^+ + \boldsymbol{v}^-) \quad , \quad \boldsymbol{\psi}_2 = c_2 (\boldsymbol{v}^+ - \boldsymbol{v}^-), \tag{53}$$

where $c_1$ and $c_2$ are normalization constants. We prove that $\boldsymbol{\psi}_1$ is an eigenvector of $\boldsymbol{C}$; similar argument applies to $\boldsymbol{\psi}_2$.

$$\begin{aligned}
\boldsymbol{C\psi}_1 &= (\boldsymbol{v}^+\boldsymbol{v}^{+^T} + \boldsymbol{v}^-\boldsymbol{v}^{-^T}) c_1 (\boldsymbol{v}^+ + \boldsymbol{v}^-) && (54) \\
&= c_1 \left( \boldsymbol{v}^+\|\boldsymbol{v}^+\|^2 + \boldsymbol{v}^-\langle \boldsymbol{v}^+, \boldsymbol{v}^-\rangle + \boldsymbol{v}^+\langle \boldsymbol{v}^+, \boldsymbol{v}^-\rangle + \boldsymbol{v}^-\|\boldsymbol{v}^-\|^2 \right) && (55) \\
&= c_1 \left( \boldsymbol{v}^+\|\boldsymbol{v}^+\|^2 + \boldsymbol{v}^-\langle \boldsymbol{v}^+, \boldsymbol{v}^-\rangle + \boldsymbol{v}^+\langle \boldsymbol{v}^+, \boldsymbol{v}^-\rangle + \boldsymbol{v}^-\|\boldsymbol{v}^-\|^2 \right) && (56) \\
&= c_1 (\|\boldsymbol{v}^+\|^2 + \langle \boldsymbol{v}^+, \boldsymbol{v}^-\rangle)\boldsymbol{v}^+ + c_1 (\|\boldsymbol{v}^-\|^2 + \langle \boldsymbol{v}^+, \boldsymbol{v}^-\rangle)\boldsymbol{v}^- && (57) \\
&= c_1 (\|\boldsymbol{v}^+\|^2 + \langle \boldsymbol{v}^+, \boldsymbol{v}^-\rangle)\boldsymbol{v}^+ + c_1 (\|\boldsymbol{v}^+\|^2 + \langle \boldsymbol{v}^+, \boldsymbol{v}^-\rangle)\boldsymbol{v}^- && (58) \\
&= (\|\boldsymbol{v}^+\|^2 + \langle \boldsymbol{v}^+, \boldsymbol{v}^-\rangle) c_1 (\boldsymbol{v}_1 + \boldsymbol{v}_2) && (59) \\
&= (\|\boldsymbol{v}^+\|^2 + \langle \boldsymbol{v}^+, \boldsymbol{v}^-\rangle) \boldsymbol{\psi}_1 . && (60)
\end{aligned}$$

The above shows that $\boldsymbol{\psi}_1$ satisfies $\boldsymbol{C\psi}_1 \propto \boldsymbol{\psi}_1$ and thus $\boldsymbol{\psi}_1$ must be an eigenvector of $\boldsymbol{C}$. Plugging the definition for $\boldsymbol{v}^+$ and $\boldsymbol{v}^-$ gives us the eigenvectors of our $\boldsymbol{C}$ matrix, and thus obtain the solution $\boldsymbol{a}$ in the equation (48). Since we are not constraining the length of the eigenvectors here, we can absorb all the constants in equation (48) into one and express the solution pair $\boldsymbol{a}$ as below.

$$\boldsymbol{a} \propto \frac{\boldsymbol{v}^+ + \boldsymbol{v}^-}{2} = \left[ \|\boldsymbol{A\mu}\| \quad 0 \quad 0 \quad \frac{(a_1\mu_1)^2}{\|\boldsymbol{A\mu}\|} \quad \frac{(a_2\mu_2)^2}{\|\boldsymbol{A\mu}\|} \quad \frac{\sqrt{2}a_1\mu_1 a_2\mu_2}{\|\boldsymbol{A\mu}\|} \right]^T \tag{61}$$

$$\boldsymbol{a} \propto \frac{\boldsymbol{v}^+ - \boldsymbol{v}^-}{2} = \left[ 0 \quad \sqrt{2}a_1\mu_1 \quad \sqrt{2}a_2\mu_2 \quad 0 \quad 0 \quad 0 \right]^T . \tag{62}$$

Recall from (32) that,

$$\phi(\boldsymbol{x}) = \frac{a^*}{\lambda} \|\boldsymbol{x}\| (a_0 + a_1 \frac{\sqrt{2}x_1}{\|\boldsymbol{x}\|} + a_2 \frac{\sqrt{2}x_2}{\|\boldsymbol{x}\|} + a_3 \frac{x_1^2}{\|\boldsymbol{x}\|^2} + a_4 \frac{x_2^2}{\|\boldsymbol{x}\|^2} + a_5 \frac{\sqrt{2}x_1 x_2}{\|\boldsymbol{x}\|^2}). \tag{63}$$

Plugging the pair of solutions for $\boldsymbol{a}$ into the above yields the two eigenfunctions,

$$\phi_1(\boldsymbol{x}) \propto \|\boldsymbol{x}\| \left( \|\boldsymbol{A\mu}\| + \frac{(a_1\mu_1)^2}{\|\boldsymbol{A\mu}\|} \frac{x_1^2}{\|\boldsymbol{x}\|^2} + \frac{(a_2\mu_2)^2}{\|\boldsymbol{A\mu}\|} \frac{x_2^2}{\|\boldsymbol{x}\|^2} + \frac{\sqrt{2}a_1\mu_1 a_2\mu_2}{\|\boldsymbol{A\mu}\|} \frac{\sqrt{2}x_1 x_2}{\|\boldsymbol{x}\|^2} \right) \tag{64}$$

$$\phi_2(\boldsymbol{x}) \propto \|\boldsymbol{x}\| \left( \sqrt{2}a_1\mu_1 \frac{\sqrt{2}x_1}{\|\boldsymbol{x}\|} + \sqrt{2}a_2\mu_2 \frac{\sqrt{2}x_2}{\|\boldsymbol{x}\|} \right) , \tag{65}$$

or more simply,

$$\phi_1(\boldsymbol{x}) \propto \|\boldsymbol{x}\| \|\boldsymbol{A\mu}\| \left( 1 + \left\langle \frac{\boldsymbol{x}}{\|\boldsymbol{x}\|}, \frac{\boldsymbol{A\mu}}{\|\boldsymbol{A\mu}\|} \right\rangle^2 \right) \tag{66}$$

$$\phi_2(\boldsymbol{x}) \propto \langle \boldsymbol{x}, \boldsymbol{A\mu} \rangle , \tag{67}$$

We convert the $\propto$ to equality by applying proper scaling factors $d_1$ and $d_2$ as follows.

$$\phi_1(\boldsymbol{x}) = d_1 \|\boldsymbol{x}\| \|\boldsymbol{A\mu}\| \left( 1 + \left\langle \frac{\boldsymbol{x}}{\|\boldsymbol{x}\|}, \frac{\boldsymbol{A\mu}}{\|\boldsymbol{A\mu}\|} \right\rangle^2 \right) \tag{68}$$

$$\phi_2(\boldsymbol{x}) = d_2 \langle \boldsymbol{x}, \boldsymbol{A\mu} \rangle , \tag{69}$$

In order to find the factors $d_1$ and $d_2$, we require $\phi_1$ and $\phi_2$ to have unit norm in the sense of (6). Starting with $\phi_1$ we proceed as,

$$
\begin{aligned}
\|\phi_1\|^2 &= d_1^2 \|\boldsymbol{A\mu}\|^2 \int_{\mathbb{R}^2} \|\boldsymbol{x}\|^2 \left(1 + \left\langle \frac{\boldsymbol{x}}{\|\boldsymbol{x}\|}, \frac{\boldsymbol{A\mu}}{\|\boldsymbol{A\mu}\|} \right\rangle^2\right)^2 p(\boldsymbol{x}) \, d\boldsymbol{x} \tag{70} \\
&= d_1^2 \|\boldsymbol{A\mu}\|^2 \left(\frac{1}{2}\|\boldsymbol{x}\|^2 \left(1 + \left\langle \frac{\boldsymbol{x}}{\|\boldsymbol{x}\|}, \frac{\boldsymbol{A\mu}}{\|\boldsymbol{A\mu}\|} \right\rangle^2\right)^2 \Big|_{\boldsymbol{x}=\boldsymbol{A\mu}} + \frac{1}{2}\|\boldsymbol{x}\|^2 \left(1 + \left\langle \frac{\boldsymbol{x}}{\|\boldsymbol{x}\|}, \frac{\boldsymbol{A\mu}}{\|\boldsymbol{A\mu}\|} \right\rangle^2\right)^2 \Big|_{\boldsymbol{x}=-\boldsymbol{A\mu}}\right) \tag{71} \\
&= d_1^2 \|\boldsymbol{A\mu}\|^2 \|\boldsymbol{x}\|^2 \left(1 + \left\langle \frac{\boldsymbol{x}}{\|\boldsymbol{x}\|}, \frac{\boldsymbol{A\mu}}{\|\boldsymbol{A\mu}\|} \right\rangle^2\right)^2 \Big|_{\boldsymbol{x}=\boldsymbol{A\mu}} \tag{72} \\
&= d_1^2 \|\boldsymbol{A\mu}\|^4 \left(1 + \left\langle \frac{\boldsymbol{A\mu}}{\|\boldsymbol{A\mu}\|}, \frac{\boldsymbol{A\mu}}{\|\boldsymbol{A\mu}\|} \right\rangle^2\right)^2 \tag{73} \\
&= d_1^2 \|\boldsymbol{A\mu}\|^4 \left(1 + 1^2\right)^2. \tag{74}
\end{aligned}
$$

Setting the above to 1 and solving in $d_1$ yields,

$$
d_1 = \frac{1}{2\|\boldsymbol{A\mu}\|^2}, \tag{75}
$$

and consequently,

$$
\phi_1(\boldsymbol{x}) = \frac{\|\boldsymbol{x}\| \left(1 + \left\langle \frac{\boldsymbol{x}}{\|\boldsymbol{x}\|}, \frac{\boldsymbol{A\mu}}{\|\boldsymbol{A\mu}\|} \right\rangle^2\right)}{2\|\boldsymbol{A\mu}\|}. \tag{76}
$$

Similarly for $\phi_2$ we proceed as,

$$
\begin{aligned}
\|\phi_2\|^2 &= d_2^2 \int_{\mathbb{R}^2} \langle \boldsymbol{x}, \boldsymbol{A\mu} \rangle^2 \, p(\boldsymbol{x}) \, d\boldsymbol{x} \tag{77} \\
&= d_2^2 \left(\frac{1}{2}\langle \boldsymbol{x}, \boldsymbol{A\mu} \rangle^2 \Big|_{\boldsymbol{x}=\boldsymbol{A\mu}} + \frac{1}{2}\langle \boldsymbol{x}, \boldsymbol{A\mu} \rangle^2 \Big|_{\boldsymbol{x}=-\boldsymbol{A\mu}}\right) \tag{78} \\
&= d_2^2 \langle \boldsymbol{A\mu}, \boldsymbol{A\mu} \rangle^2 \tag{79} \\
&= d_2^2 \|\boldsymbol{A\mu}\|^4. \tag{80}
\end{aligned}
$$

Setting the above to 1 and solving in $d_2$ yields,

$$
d_2 = \frac{1}{\|\boldsymbol{A\mu}\|^2}, \tag{81}
$$

and consequently,

$$
\phi_2(\boldsymbol{x}) = \left\langle \frac{\boldsymbol{x}}{\|\boldsymbol{A\mu}\|}, \frac{\boldsymbol{A\mu}}{\|\boldsymbol{A\mu}\|} \right\rangle. \tag{82}
$$

## E.2 EIGENVALUES

To find the eigenvalues, we simply put the eigenfunctions in their defining equation (1). Starting with $\phi_1$ we proceed as below.

$$\int_{\mathbb{R}^2} k(\boldsymbol{x}, \boldsymbol{z})\, \phi_1(\boldsymbol{z})\, p(\boldsymbol{z})\, d\boldsymbol{z} \tag{83}$$

$$= \int_{\mathbb{R}^2} a^* \|\boldsymbol{x}\| \|\boldsymbol{z}\| \left(1 + \left\langle \frac{\boldsymbol{x}}{\|\boldsymbol{x}\|}, \frac{\boldsymbol{z}}{\|\boldsymbol{z}\|} \right\rangle\right)^2 \phi_1(\boldsymbol{z})\, p(\boldsymbol{z})\, d\boldsymbol{z} \tag{84}$$

$$= \int_{\mathbb{R}^2} a^* \|\boldsymbol{x}\| \|\boldsymbol{z}\| \left(1 + \left\langle \frac{\boldsymbol{x}}{\|\boldsymbol{x}\|}, \frac{\boldsymbol{z}}{\|\boldsymbol{z}\|} \right\rangle\right)^2 \frac{\|\boldsymbol{z}\|}{\|\boldsymbol{A}\boldsymbol{\mu}\|} \frac{1 + \left\langle \frac{\boldsymbol{z}}{\|\boldsymbol{z}\|}, \frac{\boldsymbol{A}\boldsymbol{\mu}}{\|\boldsymbol{A}\boldsymbol{\mu}\|} \right\rangle^2}{2}\, p(\boldsymbol{z})\, d\boldsymbol{z} \tag{85}$$

$$= \frac{1}{2}\left( a^* \|\boldsymbol{x}\| \|\boldsymbol{z}\| \left(1 + \left\langle \frac{\boldsymbol{x}}{\|\boldsymbol{x}\|}, \frac{\boldsymbol{z}}{\|\boldsymbol{z}\|} \right\rangle\right)^2 \frac{\|\boldsymbol{z}\|}{\|\boldsymbol{A}\boldsymbol{\mu}\|} \frac{1 + \left\langle \frac{\boldsymbol{z}}{\|\boldsymbol{z}\|}, \frac{\boldsymbol{A}\boldsymbol{\mu}}{\|\boldsymbol{A}\boldsymbol{\mu}\|} \right\rangle^2}{2} \right)_{\boldsymbol{z}=\boldsymbol{A}\boldsymbol{\mu}} \tag{86}$$

$$+ \frac{1}{2}\left( a^* \|\boldsymbol{x}\| \|\boldsymbol{z}\| \left(1 + \left\langle \frac{\boldsymbol{x}}{\|\boldsymbol{x}\|}, \frac{\boldsymbol{z}}{\|\boldsymbol{z}\|} \right\rangle\right)^2 \frac{\|\boldsymbol{z}\|}{\|\boldsymbol{A}\boldsymbol{\mu}\|} \frac{1 + \left\langle \frac{\boldsymbol{z}}{\|\boldsymbol{z}\|}, \frac{\boldsymbol{A}\boldsymbol{\mu}}{\|\boldsymbol{A}\boldsymbol{\mu}\|} \right\rangle^2}{2} \right)_{\boldsymbol{z}=-\boldsymbol{A}\boldsymbol{\mu}} \tag{87}$$

$$= \frac{1}{2}\left( a^* \|\boldsymbol{x}\| \|\boldsymbol{A}\boldsymbol{\mu}\| \left(1 + \left\langle \frac{\boldsymbol{x}}{\|\boldsymbol{x}\|}, \frac{\boldsymbol{A}\boldsymbol{\mu}}{\|\boldsymbol{A}\boldsymbol{\mu}\|} \right\rangle\right)^2 \frac{1 + \left\langle \frac{\boldsymbol{A}\boldsymbol{\mu}}{\|\boldsymbol{A}\boldsymbol{\mu}\|}, \frac{\boldsymbol{A}\boldsymbol{\mu}}{\|\boldsymbol{A}\boldsymbol{\mu}\|} \right\rangle^2}{2} \right) \tag{88}$$

$$+ \frac{1}{2}\left( a^* \|\boldsymbol{x}\| \|-\boldsymbol{A}\boldsymbol{\mu}\| \left(1 + \left\langle \frac{\boldsymbol{x}}{\|\boldsymbol{x}\|}, \frac{-\boldsymbol{A}\boldsymbol{\mu}}{\|-\boldsymbol{A}\boldsymbol{\mu}\|} \right\rangle\right)^2 \frac{1 + \left\langle \frac{-\boldsymbol{A}\boldsymbol{\mu}}{\|-\boldsymbol{A}\boldsymbol{\mu}\|}, \frac{\boldsymbol{A}\boldsymbol{\mu}}{\|\boldsymbol{A}\boldsymbol{\mu}\|} \right\rangle^2}{2} \right) \tag{89}$$

$$= \frac{1}{2} a^* \|\boldsymbol{x}\| \|\boldsymbol{A}\boldsymbol{\mu}\| \left( \left(1 + \left\langle \frac{\boldsymbol{x}}{\|\boldsymbol{x}\|}, \frac{\boldsymbol{A}\boldsymbol{\mu}}{\|\boldsymbol{A}\boldsymbol{\mu}\|} \right\rangle\right)^2 + \left(1 + \left\langle \frac{\boldsymbol{x}}{\|\boldsymbol{x}\|}, \frac{-\boldsymbol{A}\boldsymbol{\mu}}{\|-\boldsymbol{A}\boldsymbol{\mu}\|} \right\rangle\right)^2 \right) \tag{90}$$

$$= a^* \|\boldsymbol{x}\| \|\boldsymbol{A}\boldsymbol{\mu}\| \left(1 + \left\langle \frac{\boldsymbol{x}}{\|\boldsymbol{x}\|}, \frac{\boldsymbol{A}\boldsymbol{\mu}}{\|\boldsymbol{A}\boldsymbol{\mu}\|} \right\rangle^2 \right) \tag{91}$$

$$= 2a^* \|\boldsymbol{A}\boldsymbol{\mu}\|^2 \frac{\|\boldsymbol{x}\|}{2\|\boldsymbol{A}\boldsymbol{\mu}\|} \left(1 + \left\langle \frac{\boldsymbol{x}}{\|\boldsymbol{x}\|}, \frac{\boldsymbol{A}\boldsymbol{\mu}}{\|\boldsymbol{A}\boldsymbol{\mu}\|} \right\rangle^2 \right) \tag{92}$$

$$= 2a^* \|\boldsymbol{A}\boldsymbol{\mu}\|^2\, \phi_1(\boldsymbol{x})\,. \tag{93}$$

Therefore,

$$\lambda_1 = 2a^* \|\boldsymbol{A}\boldsymbol{\mu}\|^2\,. \tag{94}$$

Similarly for $\phi_2$ we have,

$$\int_{\mathbb{R}^2} k(\boldsymbol{x}, \boldsymbol{z})\, \phi_2(\boldsymbol{z})\, p(\boldsymbol{z})\, d\boldsymbol{z} \tag{95}$$

$$= \int_{\mathbb{R}^2} a^* \|\boldsymbol{x}\| \|\boldsymbol{z}\| \left(1 + \left\langle \frac{\boldsymbol{x}}{\|\boldsymbol{x}\|}, \frac{\boldsymbol{z}}{\|\boldsymbol{z}\|} \right\rangle\right)^2 \phi_2(\boldsymbol{z})\, p(\boldsymbol{z})\, d\boldsymbol{z} \tag{96}$$

$$= \int_{\mathbb{R}^2} a^* \|\boldsymbol{x}\| \|\boldsymbol{z}\| \left(1 + \left\langle \frac{\boldsymbol{x}}{\|\boldsymbol{x}\|}, \frac{\boldsymbol{z}}{\|\boldsymbol{z}\|} \right\rangle\right)^2 \left\langle \frac{\boldsymbol{z}}{\|\boldsymbol{A\mu}\|}, \frac{\boldsymbol{A\mu}}{\|\boldsymbol{A\mu}\|} \right\rangle p(\boldsymbol{z})\, d\boldsymbol{z} \tag{97}$$

$$= \frac{1}{2}\left( a^* \|\boldsymbol{x}\| \|\boldsymbol{z}\| \left(1 + \left\langle \frac{\boldsymbol{x}}{\|\boldsymbol{x}\|}, \frac{\boldsymbol{z}}{\|\boldsymbol{z}\|} \right\rangle\right)^2 \left\langle \frac{\boldsymbol{z}}{\|\boldsymbol{A\mu}\|}, \frac{\boldsymbol{A\mu}}{\|\boldsymbol{A\mu}\|} \right\rangle \right)_{\boldsymbol{z} = \boldsymbol{A\mu}} \tag{98}$$

$$+ \frac{1}{2}\left( a^* \|\boldsymbol{x}\| \|\boldsymbol{z}\| \left(1 + \left\langle \frac{\boldsymbol{x}}{\|\boldsymbol{x}\|}, \frac{\boldsymbol{z}}{\|\boldsymbol{z}\|} \right\rangle\right)^2 \left\langle \frac{\boldsymbol{z}}{\|\boldsymbol{A\mu}\|}, \frac{\boldsymbol{A\mu}}{\|\boldsymbol{A\mu}\|} \right\rangle \right)_{\boldsymbol{z} = -\boldsymbol{A\mu}} \tag{99}$$

$$= \frac{1}{2}\left( a^* \|\boldsymbol{x}\| \|\boldsymbol{A\mu}\| \left(1 + \left\langle \frac{\boldsymbol{x}}{\|\boldsymbol{x}\|}, \frac{\boldsymbol{A\mu}}{\|\boldsymbol{A\mu}\|} \right\rangle\right)^2 \left\langle \frac{\boldsymbol{A\mu}}{\|\boldsymbol{A\mu}\|}, \frac{\boldsymbol{A\mu}}{\|\boldsymbol{A\mu}\|} \right\rangle \right) \tag{100}$$

$$+ \frac{1}{2}\left( a^* \|\boldsymbol{x}\| \|-\boldsymbol{A\mu}\| \left(1 + \left\langle \frac{\boldsymbol{x}}{\|\boldsymbol{x}\|}, \frac{-\boldsymbol{A\mu}}{\|-\boldsymbol{A\mu}\|} \right\rangle\right)^2 \left\langle \frac{-\boldsymbol{A\mu}}{\|-\boldsymbol{A\mu}\|}, \frac{\boldsymbol{A\mu}}{\|\boldsymbol{A\mu}\|} \right\rangle \right) \tag{101}$$

$$= \frac{1}{2}\left( a^* \|\boldsymbol{x}\| \|\boldsymbol{A\mu}\| \left(1 + \left\langle \frac{\boldsymbol{x}}{\|\boldsymbol{x}\|}, \frac{\boldsymbol{A\mu}}{\|\boldsymbol{A\mu}\|} \right\rangle\right)^2 \right) \tag{102}$$

$$+ \frac{1}{2}\left( -a^* \|\boldsymbol{x}\| \|-\boldsymbol{A\mu}\| \left(1 + \left\langle \frac{\boldsymbol{x}}{\|\boldsymbol{x}\|}, \frac{-\boldsymbol{A\mu}}{\|-\boldsymbol{A\mu}\|} \right\rangle\right)^2 \right) \tag{103}$$

$$= \frac{1}{2} a^* \|\boldsymbol{x}\| \|\boldsymbol{A\mu}\| \left( \left(1 + \left\langle \frac{\boldsymbol{x}}{\|\boldsymbol{x}\|}, \frac{\boldsymbol{A\mu}}{\|\boldsymbol{A\mu}\|} \right\rangle\right)^2 - \left(1 - \left\langle \frac{\boldsymbol{x}}{\|\boldsymbol{x}\|}, \frac{\boldsymbol{A\mu}}{\|\boldsymbol{A\mu}\|} \right\rangle\right)^2 \right) \tag{104}$$

$$= \frac{1}{2} a^* \|\boldsymbol{x}\| \|\boldsymbol{A\mu}\| \left( 4 \left\langle \frac{\boldsymbol{x}}{\|\boldsymbol{x}\|}, \frac{\boldsymbol{A\mu}}{\|\boldsymbol{A\mu}\|} \right\rangle \right) \tag{105}$$

$$= 2 a^* \|\boldsymbol{A\mu}\|^2 \left\langle \frac{\boldsymbol{x}}{\|\boldsymbol{A\mu}\|}, \frac{\boldsymbol{A\mu}}{\|\boldsymbol{A\mu}\|} \right\rangle \tag{106}$$

$$= 2 a^* \|\boldsymbol{A\mu}\|^2\, \phi_2(\boldsymbol{x})\,. \tag{107}$$

Therefore,

$$\lambda_2 = 2 a^* \|\boldsymbol{A\mu}\|^2\,. \tag{108}$$

## F  PROOF OF LINEAR KERNEL THEOREM

An eigenfunction $\phi$ of the kernel operator associated with kernel $k(\boldsymbol{x}, \boldsymbol{z}) = \langle \boldsymbol{x}, \boldsymbol{z} \rangle$ must satisfy,

$$\int_{\mathbb{R}^n} \langle \boldsymbol{x}, \boldsymbol{z} \rangle\, \phi(\boldsymbol{z}) p(\boldsymbol{z})\, d\boldsymbol{z} = \lambda\, \phi(\boldsymbol{x}) \tag{109}$$

Following our earlier setup, we proceed with a 2-dimensional input this becomes,

$$\int_{\mathbb{R}^n} \langle \boldsymbol{x}, \boldsymbol{z} \rangle\, \phi(\boldsymbol{z}) p(\boldsymbol{z})\, d\boldsymbol{z} \tag{110}$$

$$= \int_{\mathbb{R}^2} (x_1 z_1 + x_2 z_2)\, \phi(\boldsymbol{z}) p(\boldsymbol{z})\, d\boldsymbol{z} \tag{111}$$

$$= a_1 x_1 + a_2 x_2\,, \tag{112}$$

where $a_1$ and $a_2$ are the result of the definite integrals of form $\int_{\mathbb{R}^2} x_i f(\boldsymbol{z})\, d\boldsymbol{z}$, for $i = 1, 2$. Observe that each $a_i$ is thus constant in $\boldsymbol{x}$ and $\boldsymbol{z}$. Plugging the above expression into (109) yields,

$$a_1 x_1 + a_2 x_2 = \lambda\, \phi(\boldsymbol{x})\,, \tag{113}$$

Thus the eigenfunction has the form,

$$\phi(\boldsymbol{x}) \;=\; \frac{1}{\lambda}\left(a_1 x_1 + a_2 x_2\right). \tag{114}$$

In order to figure out the values of $a_1$ and $a_2$ we plug the obtained eigenfunction form into (109) again, for both $\phi(\boldsymbol{x})$ and $\phi(\boldsymbol{z})$,

$$\int_{\mathbb{R}^2} \langle \boldsymbol{x}, \boldsymbol{z} \rangle \, \phi(\boldsymbol{z}) p(\boldsymbol{z}) \, d\boldsymbol{z} \;=\; \lambda \, \phi(\boldsymbol{x}) \tag{115}$$

$$\Rightarrow \int_{\mathbb{R}^2} (x_1 z_1 + x_2 z_2) \, \frac{1}{\lambda}(a_1 z_1 + a_2 z_2) \, p(\boldsymbol{z}) \, d\boldsymbol{z} \;=\; \lambda \, \frac{1}{\lambda}(a_1 x_1 + a_2 x_2) \tag{116}$$

$$\Rightarrow \int_{\mathbb{R}^2} (x_1 z_1 + x_2 z_2) \, (a_1 z_1 + a_2 z_2) \, p(\boldsymbol{z}) \, d\boldsymbol{z} \;=\; \lambda \, (a_1 x_1 + a_2 x_2) \tag{117}$$

$$\Rightarrow \sum_i \pi_i (x_1 \mu_1^{(i)} + x_2 \mu_2^{(i)}) \, (a_1 \mu_1^{(i)} + a_2 \mu_2^{(i)}) \;=\; \lambda \, (a_1 x_1 + a_2 x_2), \tag{118}$$

where in the last line we use the assumption that the covariance matrix of each Gaussian source is small, and thus we can resort to a Laplace-like approximation of the integral. To have the above identity hold true at any $\boldsymbol{x}$ we need to require that,

$$\sum_i \pi_i \mu_1^{(i)} \, (a_1 \mu_1^{(i)} + a_2 \mu_2^{(i)}) \;=\; \lambda \, a_1 \tag{119}$$

$$\sum_i \pi_i \mu_2^{(i)} \, (a_1 \mu_1^{(i)} + a_2 \mu_2^{(i)}) \;=\; \lambda \, a_2 \tag{120}$$

$$\tag{121}$$

or in matrix form,

$$\underbrace{\left( \sum_i \pi_i \underbrace{\begin{bmatrix} {\mu_1^{(i)}}^2 & \mu_1^{(i)} \mu_2^{(i)} \\ \mu_1^{(i)} \mu_2^{(i)} & {\mu_2^{(i)}}^2 \end{bmatrix}}_{\boldsymbol{C}_i} \right)}_{\boldsymbol{C}} \begin{bmatrix} a_1 \\ a_2 \end{bmatrix} = \lambda \begin{bmatrix} a_1 \\ a_2 \end{bmatrix} \tag{122}$$

The equation can be compactly expressed as,

$$\boldsymbol{C}\boldsymbol{a} = \lambda \boldsymbol{a}. \tag{123}$$

Thus, any solution $\boldsymbol{a}$ must be an eigenvector of $\boldsymbol{C}$. In the sequel we discuss how to obtain the eigenvectors of $\boldsymbol{C}$. Observe that each matrix $\boldsymbol{C}_i$ is a ***rank-one symmetric matrix*** that can be written as $\boldsymbol{\mu}^{(i)} {\boldsymbol{\mu}^{(i)}}^T$. In particular, in our 2-d problem with 2 Gaussian mixtures, where $\pi_1 = \pi_2 = 1/2$ and $\boldsymbol{\mu}^{(1)} = \boldsymbol{A}\boldsymbol{\mu}$ and $\boldsymbol{\mu}^{(2)} = -\boldsymbol{A}\boldsymbol{\mu}$, the above equation can be expressed as below.

$$\left( \frac{1}{2}(\boldsymbol{A}\boldsymbol{\mu})(\boldsymbol{A}\boldsymbol{\mu})^T + \frac{1}{2}(-\boldsymbol{A}\boldsymbol{\mu})(-\boldsymbol{A}\boldsymbol{\mu})^T \right) \boldsymbol{a} \;=\; \lambda \boldsymbol{a} \tag{124}$$

$$\Rightarrow \; (\boldsymbol{A}\boldsymbol{\mu})(\boldsymbol{A}\boldsymbol{\mu})^T \boldsymbol{a} \;=\; \lambda \boldsymbol{a}, \tag{125}$$

Thus,

$$\boldsymbol{a} \;\propto\; \boldsymbol{A}\boldsymbol{\mu}. \tag{126}$$

Recall from (114) that,

$$\phi(\boldsymbol{x}) \;=\; \frac{1}{\lambda} \langle \boldsymbol{a}, \boldsymbol{x} \rangle. \tag{127}$$

Plugging the solution for $\boldsymbol{a}$ into the above yields the eigenfunction,

$$\phi(\boldsymbol{x}) \;\propto\; \langle \boldsymbol{A}\boldsymbol{\mu}, \boldsymbol{x} \rangle, \tag{128}$$

We convert the $\propto$ to equality by applying proper scaling factors $c$ as follows.

$$\phi(\boldsymbol{x}) \;=\; c \, \langle \boldsymbol{A}\boldsymbol{\mu}, \boldsymbol{x} \rangle, \tag{129}$$

In order to find the scale factor $c$, we require $\phi$ to have unit norm in the sense of (6).

$$\|\phi\|^2 = c^2 \int_{\mathbb{R}^2} \left(\langle \boldsymbol{A\mu}, \boldsymbol{x}\rangle\right)^2 p(\boldsymbol{x})\, d\boldsymbol{x} \tag{130}$$

$$= \frac{c^2}{2} \left(\langle \boldsymbol{A\mu}, \boldsymbol{x}\rangle\right)^2_{\boldsymbol{x}=\boldsymbol{A\mu}} + \frac{c^2}{2} \left(\langle \boldsymbol{A\mu}, \boldsymbol{x}\rangle\right)^2_{\boldsymbol{x}=-\boldsymbol{A\mu}} \tag{131}$$

$$= c^2 \left(\langle \boldsymbol{A\mu}, \boldsymbol{x}\rangle\right)^2_{\boldsymbol{x}=\boldsymbol{A\mu}} \tag{132}$$

$$= c^2 \left(\|\boldsymbol{A\mu}\|^2\right)^2. \tag{133}$$

Setting the above to 1 and solving in $c$ yields,

$$c = \frac{1}{\|\boldsymbol{A\mu}\|^2}, \tag{134}$$

and consequently,

$$\phi(\boldsymbol{x}) = \frac{\langle \boldsymbol{A\mu}, \boldsymbol{x}\rangle}{\|\boldsymbol{A\mu}\|^2}. \tag{135}$$

For finding the eigenvalue associated with $\phi$ we plug this form into (109) for both $\phi(\boldsymbol{x})$ and $\phi(\boldsymbol{z})$ and then solve in $\lambda$.

$$\int_{\mathbb{R}^n} \langle \boldsymbol{x}, \boldsymbol{z}\rangle\, \phi(\boldsymbol{z}) p(\boldsymbol{z})\, d\boldsymbol{z} = \lambda\, \phi(\boldsymbol{x}) \tag{136}$$

$$\Rightarrow \int_{\mathbb{R}^n} \langle \boldsymbol{x}, \boldsymbol{z}\rangle \frac{\langle \boldsymbol{A\mu}, \boldsymbol{z}\rangle}{\|\boldsymbol{A\mu}\|^2} p(\boldsymbol{z})\, d\boldsymbol{z} = \lambda \frac{\langle \boldsymbol{A\mu}, \boldsymbol{x}\rangle}{\|\boldsymbol{A\mu}\|^2} \tag{137}$$

$$\Rightarrow \frac{1}{2}\left(\langle \boldsymbol{x}, \boldsymbol{z}\rangle \frac{\langle \boldsymbol{A\mu}, \boldsymbol{z}\rangle}{\|\boldsymbol{A\mu}\|^2}\right)_{\boldsymbol{z}=\boldsymbol{A\mu}} + \frac{1}{2}\left(\langle \boldsymbol{x}, \boldsymbol{z}\rangle \frac{\langle \boldsymbol{A\mu}, \boldsymbol{z}\rangle}{\|\boldsymbol{A\mu}\|^2}\right)_{\boldsymbol{z}=-\boldsymbol{A\mu}} = \lambda \frac{\langle \boldsymbol{A\mu}, \boldsymbol{x}\rangle}{\|\boldsymbol{A\mu}\|^2} \tag{138}$$

$$\Rightarrow \langle \boldsymbol{x}, \boldsymbol{A\mu}\rangle \frac{\langle \boldsymbol{A\mu}, \boldsymbol{A\mu}\rangle}{\|\boldsymbol{A\mu}\|^2} = \lambda \frac{\langle \boldsymbol{A\mu}, \boldsymbol{x}\rangle}{\|\boldsymbol{A\mu}\|^2} \tag{139}$$

$$\Rightarrow \|\boldsymbol{A\mu}\|^2 = \lambda. \tag{140}$$

## G  OPTIMAL PREDICTION

Consider the task of finding a function $f(\boldsymbol{x})$ that best approximates a given function $y(\boldsymbol{x})$ via the following regression setup.

$$L = \frac{1}{2} \int_{\mathbb{R}^n} \left(f(\boldsymbol{x}) - y(\boldsymbol{x})\right)^2 p(\boldsymbol{x}) d\boldsymbol{x} \tag{141}$$

Suppose we are interested in expressing $f$ using the bases induced by the kernel $k$, that is the eigenfunctions of the linear operator associated with $k$ whose eigenvaules are non-zero. That is,

$$f(\boldsymbol{x}) = \sum_k a_k \phi_k(\boldsymbol{x}), \tag{142}$$

where $\phi_k$ is an eigenfunction, and $k$ runs over eigenfunctions with non-zero eigenvalue. We now show that the function $f$ which minimizes the regression loss $L$ has the form,

$$f(\boldsymbol{x}) = \sum_k a_k \phi_k(\boldsymbol{x}) \tag{143}$$

$$= \sum_k \int_{\mathbb{R}^n} \phi_k(\boldsymbol{z}) y(\boldsymbol{z})\, p(\boldsymbol{z}) d\boldsymbol{z} \phi_k(\boldsymbol{x}). \tag{144}$$

We start from,

$$L = \frac{1}{2} \int_{\mathbb{R}^n} \left(f(\boldsymbol{x}) - y(\boldsymbol{x})\right)^2 p(\boldsymbol{x}) d\boldsymbol{x} \tag{145}$$

$$= \frac{1}{2} \int_{\mathbb{R}^n} \left(\sum_k a_k \phi_k(\boldsymbol{x}) - y(\boldsymbol{x})\right)^2 p(\boldsymbol{x}) d\boldsymbol{x}. \tag{146}$$

Thus,

$$\frac{\partial}{\partial a_j} L = \int_{\mathbb{R}^n} \Big( \sum_k a_k \phi_k(\boldsymbol{x}) - y(\boldsymbol{x}) \Big) \Big( a_j \phi_j(\boldsymbol{x}) \Big) p(\boldsymbol{x}) d\boldsymbol{x} \tag{147}$$

$$= \int_{\mathbb{R}^n} \Big( \sum_k a_k a_j \phi_k(\boldsymbol{x}) \phi_j(\boldsymbol{x}) - a_j \phi_j(\boldsymbol{x}) y(\boldsymbol{x}) \Big) p(\boldsymbol{x}) d\boldsymbol{x} \tag{148}$$

$$= \Big( \sum_k a_k a_j \int_{\mathbb{R}^n} \phi_k(\boldsymbol{x}) \phi_j(\boldsymbol{x}) \, p(\boldsymbol{x}) d\boldsymbol{x} \Big) - a_j \int_{\mathbb{R}^n} \phi_j(\boldsymbol{x}) y(\boldsymbol{x}) \, p(\boldsymbol{x}) d\boldsymbol{x} \tag{149}$$

$$= a_j^2 - a_j \int_{\mathbb{R}^n} \phi_j(\boldsymbol{x}) y(\boldsymbol{x}) \, p(\boldsymbol{x}) d\boldsymbol{x} \,. \tag{150}$$

Zero-crossing,

$$a_j = \int_{\mathbb{R}^n} \phi_j(\boldsymbol{x}) y(\boldsymbol{x}) \, p(\boldsymbol{x}) d\boldsymbol{x} \,. \tag{151}$$

Thus,

$$f(\boldsymbol{x}) = \sum_k a_k \phi_k(\boldsymbol{x}) \tag{152}$$

$$= \sum_k \int_{\mathbb{R}^n} \phi_k(\boldsymbol{z}) y(\boldsymbol{z}) \, p(\boldsymbol{z}) d\boldsymbol{z} \phi_k(\boldsymbol{x}) \tag{153}$$

In particular, if we replace $p(\boldsymbol{z})$ by our pair of Gaussian densities with small covariance, and set $y(\boldsymbol{x})$ to 1 and 0 depending on whether $\boldsymbol{x}$ is drawn from the positive or negative component of the mixture, we arrive at,

$$f(\boldsymbol{x}) = \sum_i \Big( \phi_i(\boldsymbol{x}) \big( \frac{1}{2} \int_{\mathbb{R}^2} \phi_i(\boldsymbol{z}) \, (1) \, p^+(\boldsymbol{z}) \, d\boldsymbol{z} + \frac{1}{2} \int_{\mathbb{R}^2} \phi_i(\boldsymbol{z}) \, (0) \, p^-(\boldsymbol{z}) \, d\boldsymbol{z} \big) \Big) = \frac{1}{2} \sum_i \phi_i(\boldsymbol{x}) \phi_i(\boldsymbol{A\mu}) \,, \tag{154}$$

where $p^+$ and $p^-$ denote class-conditional normal distributions.

## H  SENSITIVITY ANALYSIS

### H.1  LINEAR KERNEL

**Theorem 5** *In a linear network, $|\zeta_1| - |\zeta_2|$ is always zero for any choice of $m \geq 1$ and regardless of the values of $a_1$ and $a_2$.*

Recall the definitions,

$$f(\boldsymbol{x}) = \frac{1}{2} \sum_i \phi_i(\boldsymbol{x}) \phi_i(\boldsymbol{A\mu}) \tag{155}$$

$$g_{\boldsymbol{B}}(\boldsymbol{x}) \triangleq \frac{1}{2} \sum_i \phi_i(\boldsymbol{x}) \phi_i(\boldsymbol{BA\mu}) \tag{156}$$

$$\gamma(\boldsymbol{b}) \triangleq \int_{\mathbb{R}^2} \frac{f(\boldsymbol{x})}{\sqrt{\int_{\mathbb{R}^2} f^2(\boldsymbol{t}) \, p(\boldsymbol{t}) \, d\boldsymbol{t}}} \frac{g_{\boldsymbol{B}}(\boldsymbol{x})}{\sqrt{\int_{\mathbb{R}^2} g_{\boldsymbol{B}}^2(\boldsymbol{t}) \, p(\boldsymbol{t}) \, d\boldsymbol{t}}} p(\boldsymbol{x}) \, d\boldsymbol{x} \,. \tag{157}$$

For linear kernel, we had only one eigenfunction with the form,

$$\phi_1(\boldsymbol{x}) = \frac{\langle \boldsymbol{A\mu}, \boldsymbol{x} \rangle}{\|\boldsymbol{A\mu}\|^2} \,. \tag{158}$$

Replacing that, in the above definitions yields,

$$f(\boldsymbol{x}) = \frac{1}{2} \frac{\langle \boldsymbol{A\mu}, \boldsymbol{x} \rangle}{\|\boldsymbol{A\mu}\|^2} \frac{\langle \boldsymbol{A\mu}, \boldsymbol{A\mu} \rangle}{\|\boldsymbol{A\mu}\|^2} = \frac{1}{2} \frac{\langle \boldsymbol{A\mu}, \boldsymbol{x} \rangle}{\|\boldsymbol{A\mu}\|^2} \tag{159}$$

$$g_{\boldsymbol{B}}(\boldsymbol{x}) = \frac{1}{2} \frac{\langle \boldsymbol{A\mu}, \boldsymbol{x} \rangle}{\|\boldsymbol{A\mu}\|^2} \frac{\langle \boldsymbol{A\mu}, \boldsymbol{BA\mu} \rangle}{\|\boldsymbol{A\mu}\|^2} \,. \tag{160}$$

It is easy to obtain,

$$\int_{\mathbb{R}^n} f^2(\boldsymbol{x}) p(\boldsymbol{x}) \quad = \quad \frac{1}{4} \frac{1}{\|\boldsymbol{A\mu}\|^4} \int_{\mathbb{R}^n} (\langle \boldsymbol{A\mu}, \boldsymbol{x} \rangle)^2 \, p(\boldsymbol{x}) \, d\boldsymbol{x} \tag{161}$$

$$= \quad \frac{1}{4} \frac{1}{\|\boldsymbol{A\mu}\|^4} \, 2(\langle \boldsymbol{A\mu}, \boldsymbol{A\mu} \rangle)^2 \ = \ \frac{1}{2} \,, \tag{162}$$

and,

$$\int_{\mathbb{R}^n} g^2(\boldsymbol{x}) p(\boldsymbol{x}) \quad = \quad \frac{1}{4} \frac{(\langle \boldsymbol{A\mu}, \boldsymbol{BA\mu} \rangle)^2}{\|\boldsymbol{A\mu}\|^8} \int_{\mathbb{R}^n} (\langle \boldsymbol{A\mu}, \boldsymbol{x} \rangle)^2 \, p(\boldsymbol{x}) \, d\boldsymbol{x} \tag{163}$$

$$= \quad \frac{1}{4} \frac{(\langle \boldsymbol{A\mu}, \boldsymbol{BA\mu} \rangle)^2}{\|\boldsymbol{A\mu}\|^8} \, 2(\langle \boldsymbol{A\mu}, \boldsymbol{A\mu} \rangle)^2 \tag{164}$$

$$= \quad \frac{1}{2} \frac{(\langle \boldsymbol{A\mu}, \boldsymbol{BA\mu} \rangle)^2}{\|\boldsymbol{A\mu}\|^4} \,. \tag{165}$$

Plugging these results into the definition of $\gamma$ yields,

$$\gamma(\boldsymbol{b}) \quad \triangleq \quad \int_{\mathbb{R}^2} \frac{f(\boldsymbol{x})}{\sqrt{\int_{\mathbb{R}^2} f^2(\boldsymbol{t}) \, p(\boldsymbol{t}) \, d\boldsymbol{t}}} \frac{g_B(\boldsymbol{x})}{\sqrt{\int_{\mathbb{R}^2} g_B^2(\boldsymbol{t}) \, p(\boldsymbol{t}) \, d\boldsymbol{t}}} \, p(\boldsymbol{x}) \, d\boldsymbol{x} \tag{166}$$

$$= \quad \int_{\mathbb{R}^2} \frac{\frac{1}{2} \frac{\langle \boldsymbol{A\mu}, \boldsymbol{x} \rangle}{\|\boldsymbol{A\mu}\|^2}}{\frac{1}{\sqrt{2}}} \frac{\frac{1}{2} \frac{\langle \boldsymbol{A\mu}, \boldsymbol{x} \rangle}{\|\boldsymbol{A\mu}\|^2} \frac{\langle \boldsymbol{A\mu}, \boldsymbol{BA\mu} \rangle}{\|\boldsymbol{A\mu}\|^2}}{\frac{1}{\sqrt{2}} \frac{|\langle \boldsymbol{A\mu}, \boldsymbol{BA\mu} \rangle|}{\|\boldsymbol{A\mu}\|^2}} \, p(\boldsymbol{x}) \, d\boldsymbol{x} \tag{167}$$

$$= \quad \int_{\mathbb{R}^2} \frac{\frac{1}{2} \frac{\langle \boldsymbol{A\mu}, \boldsymbol{x} \rangle}{\|\boldsymbol{A\mu}\|^2}}{\frac{1}{\sqrt{2}}} \frac{\frac{1}{2} \frac{\langle \boldsymbol{A\mu}, \boldsymbol{x} \rangle}{\|\boldsymbol{A\mu}\|^2} \, \mathrm{sign}(\langle \boldsymbol{A\mu}, \boldsymbol{BA\mu} \rangle)}{\frac{1}{\sqrt{2}}} \, p(\boldsymbol{x}) \, d\boldsymbol{x} \tag{168}$$

$$= \quad \frac{\mathrm{sign}(\langle \boldsymbol{A\mu}, \boldsymbol{BA\mu} \rangle)}{2\|\boldsymbol{A\mu}\|^4} \int_{\mathbb{R}^2} (\langle \boldsymbol{A\mu}, \boldsymbol{x} \rangle)^2 \, p(\boldsymbol{x}) \, d\boldsymbol{x} \tag{169}$$

$$= \quad \mathrm{sign}(\langle \boldsymbol{A\mu}, \boldsymbol{BA\mu} \rangle) \tag{170}$$

$$= \quad \mathrm{sign}(b_1 a_1^2 \mu_1^2 + b_2 a_2^2 \mu_2^2) \,. \tag{171}$$

Recall that,

$$\zeta_i \quad \triangleq \quad \left( \frac{\partial^m}{\partial b_i^m} \gamma \right)_{\boldsymbol{b=1}} \,. \tag{172}$$

It is easy to see that when for any $m \geq 1$,

$$\frac{\partial^m}{\partial b_i^m} \gamma = \begin{cases} 0 & b_1 a_1^2 \mu_1^2 + b_2 a_2^2 \mu_2^2 \neq 0 \\ \mathtt{NaN} & b_1 a_1^2 \mu_1^2 + b_2 a_2^2 \mu_2^2 = 0 \end{cases} . \tag{173}$$

To find $\zeta_1$ and $\zeta_2$ we need to evaluate the above at the point $b_1 = b_2 = 1$. Since by definition, $\mu_i \neq 0$ and $a_i \neq 0$, it easy follows that $\zeta_1 = \zeta_2 = 0$ and thus $|\zeta_1| = |\zeta_2|$ for any $m \geq 0$.

## H.2   RELU LINEAR KERNEL

**Theorem 6** *In a ReLU network, $|\zeta_1| - |\zeta_2| = 0$ for any $1 \leq m \leq 8$. The first non-zero $|\zeta_1| - |\zeta_2|$ happens at $m = 9$ and has the following form,*

$$|\zeta_1| - |\zeta_2| \quad = \quad \frac{5670}{\|\boldsymbol{A\mu}\|^{18}} (a_1 a_2 \mu_1 \mu_2)^8 (a_1^2 \mu_1^2 - a_2^2 \mu_2^2) \,. \tag{174}$$

Recall the definitions,

$$f(\boldsymbol{x}) \quad = \quad \frac{1}{2} \sum_i \phi_i(\boldsymbol{x}) \phi_i(\boldsymbol{A\mu}) \tag{175}$$

$$g_B(\boldsymbol{x}) \quad \triangleq \quad \frac{1}{2} \sum_i \phi_i(\boldsymbol{x}) \phi_i(\boldsymbol{BA\mu}) \tag{176}$$

$$\gamma(\boldsymbol{b}) \quad \triangleq \quad \int_{\mathbb{R}^2} \frac{f(\boldsymbol{x})}{\sqrt{\int_{\mathbb{R}^2} f^2(\boldsymbol{t}) \, p(\boldsymbol{t}) \, d\boldsymbol{t}}} \frac{g_B(\boldsymbol{x})}{\sqrt{\int_{\mathbb{R}^2} g_B^2(\boldsymbol{t}) \, p(\boldsymbol{t}) \, d\boldsymbol{t}}} \, p(\boldsymbol{x}) \, d\boldsymbol{x} \,. \tag{177}$$

For ReLU kernel, we had only two eigenfunctions with the form,

$$\phi_1(\boldsymbol{x}) \;=\; \frac{\|\boldsymbol{x}\|}{\|\boldsymbol{A\mu}\|}\frac{1+\left\langle \frac{\boldsymbol{x}}{\|\boldsymbol{x}\|}, \frac{\boldsymbol{A\mu}}{\|\boldsymbol{A\mu}\|}\right\rangle^2}{2} \qquad , \qquad \phi_2(\boldsymbol{x}) \;=\; \left\langle \frac{\boldsymbol{x}}{\|\boldsymbol{A\mu}\|}, \frac{\boldsymbol{A\mu}}{\|\boldsymbol{A\mu}\|}\right\rangle . \tag{178}$$

Replacing them in the above definitions yields,

$$
\begin{aligned}
f(\boldsymbol{x}) &= \frac{1}{2}\frac{\|\boldsymbol{x}\|}{\|\boldsymbol{A\mu}\|}\frac{1+\left\langle \frac{\boldsymbol{x}}{\|\boldsymbol{x}\|}, \frac{\boldsymbol{A\mu}}{\|\boldsymbol{A\mu}\|}\right\rangle^2}{2}\frac{\|\boldsymbol{A\mu}\|}{\|\boldsymbol{A\mu}\|}\frac{1+\left\langle \frac{\boldsymbol{A\mu}}{\|\boldsymbol{A\mu}\|}, \frac{\boldsymbol{A\mu}}{\|\boldsymbol{A\mu}\|}\right\rangle^2}{2} + \frac{1}{2}\frac{\langle \boldsymbol{A\mu}, \boldsymbol{x}\rangle}{\|\boldsymbol{A\mu}\|^2}\frac{\langle \boldsymbol{A\mu}, \boldsymbol{A\mu}\rangle}{\|\boldsymbol{A\mu}\|^2} \tag{179} \\
&= \frac{1}{8}\frac{\|\boldsymbol{x}\|}{\|\boldsymbol{A\mu}\|}\left(1+\left\langle \frac{\boldsymbol{x}}{\|\boldsymbol{x}\|}, \frac{\boldsymbol{A\mu}}{\|\boldsymbol{A\mu}\|}\right\rangle^2\right)\left(1+\left\langle \frac{\boldsymbol{A\mu}}{\|\boldsymbol{A\mu}\|}, \frac{\boldsymbol{A\mu}}{\|\boldsymbol{A\mu}\|}\right\rangle^2\right) + \frac{1}{2}\frac{\langle \boldsymbol{A\mu}, \boldsymbol{x}\rangle}{\|\boldsymbol{A\mu}\|^2} \tag{180} \\
&= \frac{1}{8}\frac{\|\boldsymbol{x}\|}{\|\boldsymbol{A\mu}\|}\left(1+\left\langle \frac{\boldsymbol{x}}{\|\boldsymbol{x}\|}, \frac{\boldsymbol{A\mu}}{\|\boldsymbol{A\mu}\|}\right\rangle^2\right)\left(1+1\right) + \frac{1}{2}\frac{\langle \boldsymbol{A\mu}, \boldsymbol{x}\rangle}{\|\boldsymbol{A\mu}\|^2} \tag{181} \\
&= \frac{1}{4}\frac{\|\boldsymbol{x}\|}{\|\boldsymbol{A\mu}\|}\left(1+\left\langle \frac{\boldsymbol{x}}{\|\boldsymbol{x}\|}, \frac{\boldsymbol{A\mu}}{\|\boldsymbol{A\mu}\|}\right\rangle^2\right) + \frac{1}{2}\frac{\langle \boldsymbol{A\mu}, \boldsymbol{x}\rangle}{\|\boldsymbol{A\mu}\|^2} , \tag{182}
\end{aligned}
$$

and,

$$
\begin{aligned}
g_B(\boldsymbol{x}) &= \frac{1}{2}\frac{\|\boldsymbol{x}\|}{\|\boldsymbol{A\mu}\|}\frac{1+\left\langle \frac{\boldsymbol{x}}{\|\boldsymbol{x}\|}, \frac{\boldsymbol{A\mu}}{\|\boldsymbol{A\mu}\|}\right\rangle^2}{2}\frac{\|\boldsymbol{BA\mu}\|}{\|\boldsymbol{A\mu}\|}\frac{1+\left\langle \frac{\boldsymbol{BA\mu}}{\|\boldsymbol{BA\mu}\|}, \frac{\boldsymbol{A\mu}}{\|\boldsymbol{A\mu}\|}\right\rangle^2}{2} + \frac{1}{2}\frac{\langle \boldsymbol{A\mu}, \boldsymbol{x}\rangle}{\|\boldsymbol{A\mu}\|^2}\frac{\langle \boldsymbol{A\mu}, \boldsymbol{BA\mu}\rangle}{\|\boldsymbol{A\mu}\|^2} \tag{183} \\
&= \frac{1}{8}\frac{\|\boldsymbol{x}\|\|\boldsymbol{BA\mu}\|}{\|\boldsymbol{A\mu}\|^2}\left(1+\left\langle \frac{\boldsymbol{x}}{\|\boldsymbol{x}\|}, \frac{\boldsymbol{A\mu}}{\|\boldsymbol{A\mu}\|}\right\rangle^2\right)\left(1+\left\langle \frac{\boldsymbol{BA\mu}}{\|\boldsymbol{BA\mu}\|}, \frac{\boldsymbol{A\mu}}{\|\boldsymbol{A\mu}\|}\right\rangle^2\right) + \frac{1}{2}\frac{\langle \boldsymbol{A\mu}, \boldsymbol{x}\rangle}{\|\boldsymbol{A\mu}\|^2}\frac{\langle \boldsymbol{A\mu}, \boldsymbol{BA\mu}\rangle}{\|\boldsymbol{A\mu}\|^2} \tag{184}
\end{aligned}
$$

It is easy to obtain,

$$
\begin{aligned}
f^2(\boldsymbol{x}) &= \left(\frac{1}{4}\frac{\|\boldsymbol{x}\|}{\|\boldsymbol{A\mu}\|}\left(1+\left\langle \frac{\boldsymbol{x}}{\|\boldsymbol{x}\|}, \frac{\boldsymbol{A\mu}}{\|\boldsymbol{A\mu}\|}\right\rangle^2\right)\right)^2 + \left(\frac{1}{2}\frac{\langle \boldsymbol{A\mu}, \boldsymbol{x}\rangle}{\|\boldsymbol{A\mu}\|^2}\right)^2 \tag{185} \\
&\quad + 2\left(\frac{1}{4}\frac{\|\boldsymbol{x}\|}{\|\boldsymbol{A\mu}\|}\left(1+\left\langle \frac{\boldsymbol{x}}{\|\boldsymbol{x}\|}, \frac{\boldsymbol{A\mu}}{\|\boldsymbol{A\mu}\|}\right\rangle^2\right)\right)\left(\frac{1}{2}\frac{\langle \boldsymbol{A\mu}, \boldsymbol{x}\rangle}{\|\boldsymbol{A\mu}\|^2}\right) , \tag{186}
\end{aligned}
$$

and therefore,

$$
\begin{aligned}
\int_{\mathbb{R}^n} f^2(\boldsymbol{x})p(\boldsymbol{x})\,d\boldsymbol{x} &= 2\left(\frac{1}{4}\frac{\|\boldsymbol{A\mu}\|}{\|\boldsymbol{A\mu}\|}\left(1+\left\langle \frac{\boldsymbol{A\mu}}{\|\boldsymbol{A\mu}\|}, \frac{\boldsymbol{A\mu}}{\|\boldsymbol{A\mu}\|}\right\rangle^2\right)\right)^2 + 2\left(\frac{1}{2}\frac{\langle \boldsymbol{A\mu}, \boldsymbol{A\mu}\rangle}{\|\boldsymbol{A\mu}\|^2}\right)^2 \tag{187} \\
&= 2\left(\frac{1}{4}(1+1)\right)^2 + 2\left(\frac{1}{2}\right)^2 \tag{188} \\
&= 1 . \tag{189}
\end{aligned}
$$

It in a similar way, one can compute $\int_{\mathbb{R}^n} g^2(\boldsymbol{x})p(\boldsymbol{x})\,d\boldsymbol{x}$. Plugging these into the definition of $\gamma$ yields,

$$\gamma(\boldsymbol{b}) \;=\; \cfrac{1}{2\sqrt{\cfrac{2(a_1^2\mu_1^2+a_2^2\mu_2^2)^3(a_1^2b_1^2\mu_1^2+a_2^2b_2^2\mu_2^2)}{8a_1^8b_1^4\mu_1^8+8a_1^6b_1^2a_2^2(b_1+b_2)^2\mu_1^6\mu_2^2+a_1^4a_2^4(b_1+b_2)^2(b_1^2+10b_1b_2+b_2^2)\mu_1^4\mu_2^4+8a_1^2a_2^6b_2^2(b_1+b_2)^2\mu_1^2\mu_2^6+8a_2^8b_2^4\mu_2^8}}} \tag{190}$$

It is messy but straightforward to write the derivatives of $\gamma$ w.r.t. $b_1$ and $b_2$ evaluated at $b_1 = b_2 = 1$ (which gives $\zeta_1$ and $\zeta_2$) to see that for any derivative of order $m \leq 8$ the above yields $\zeta_1 = \zeta_2$ and at $m = 9$ one obtains,

$$|\zeta_1| - |\zeta_2| \;=\; \frac{5670}{\|\boldsymbol{A\mu}\|^{18}}(a_1a_2\mu_1\mu_2)^8(a_1^2\mu_1^2 - a_2^2\mu_2^2) . \tag{191}$$

## I  QUADRATIC APPROXIMATION VS RELU KERNEL

The paper presents an analytical form that characterizes the trade-off between predictivity and availability in a single-layer ReLU model for classifying two Gaussian sources.

In order to keep the theoretical analysis tractable, we the following approximations. First, we used an asymptotic approximation to the covariance matrix by having the size of the covariance going

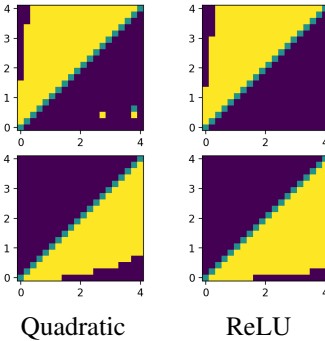

Quadratic        ReLU

Figure I.1: Plot of $\text{sign}((|\zeta_1| - |\zeta_2|)(a_1 - a_2))$ for ReLU and Quadratic NTK models trained on 50 training samples from positive $\mathcal{N}(\boldsymbol{\mu}, \boldsymbol{C})$ and negative $\mathcal{N}(-\boldsymbol{\mu}, \boldsymbol{C})$ classes. Here, $\boldsymbol{\mu} = (\mu_1, \mu_2)$, covariance is $\boldsymbol{C} = \sigma^2 \boldsymbol{I}$, where $\mu_1 = 1$ and $\sigma = 0.01$. Top and bottom figures respectively correspond to $\mu_2 = 10$ are $\mu = 0.1$.

to zero. Second, use replaced the ReLU kernel function by a quadratic approximation. A natural question is, whether the predictions made by our theory are sensitive to these approximations. More precisely, if we work with the actual ReLU kernel and a real covariance matrix, whether observe similar predictions.

We verify this here by learning a model from finite training data in the NTK regime. The latter amounts to performing a kernel regression with kernel specified by the NTK. The result of the simulation is provided in Figure I.1 and confirms that the approximations used in the theory are not affecting the actual predictions, in the sense that it matches the trend of the predictions made by the theory from Figure 5.

The left panel in Figure I.1 still uses quadratic kernel. However, it is generated by actual predictions from a model trained by finite training data, whose distribution is a real covariance (as opposed to an asymptotically small one). This confirms the closed-form expression derived from our theory (predictions from Figure 5) match the simulation result.

The right panel of Figure I.1 goes further and replaces the quadratic kernel by the actual ReLU kernel. Observe that this does not affect the trend of the predictions, hence showing there is not much lost by switching between the ReLU kernel and its quadratic approximation.

```
NUM_PLOT_POINTS = 20         # Number of points used to plot the
↪    result (larger better but slower)
N_TRAIN_SAMPLES = 50         # Number of training points per
↪    Gaussian source
USE_QUAD        = True       # When to use ReLU or its quadratic
↪    approximation
MU_1            = 1.0
MU_2_List       = [10, 0.1]
SCALE_COV       = 0.0001     # Scale parameter s from the paper
COR             = 0.0        # Abs[COR] must be less than 1
EPS             = SCALE_COV/100 # Epsilon identity added before
↪    inverting kernel matrix
DELTA           = 0.05       # delta to approximate derivatives
↪    with finite different

import numpy as np
from matplotlib import pyplot as plt

def kernel(x1,x2):
    norm_x1=np.sqrt(np.dot(x1,x1))
    norm_x2=np.sqrt(np.dot(x2,x2))
    u=np.dot(x1/norm_x1,x2/norm_x2)
```

```python
    if np.abs(u)>1.0:
        u=np.sign(u)
    if USE_QUAD:
        h=815.0/3072.0*(1 + u)**2
    else:
        h=(u*(np.pi-np.arccos(u))+np.sqrt(1-u**2))/np.pi
    return norm_x1*norm_x2*h

def kernel_predict(A,x,y,X,kernel_func):
    # do linear ridge regression to obtain coefficients c_1 to
    ↪  c_2n
    K=np.empty((2*N_TRAIN_SAMPLES,2*N_TRAIN_SAMPLES))
    for i in range(2*N_TRAIN_SAMPLES):
        for j in range(2*N_TRAIN_SAMPLES):
            K[i,j]=kernel_func(A@x[i],A@x[j])
    c = np.linalg.solve(K+EPS*np.identity(2*N_TRAIN_SAMPLES),y)

    # compute prediction vector
    K_pred=np.empty((2*N_TRAIN_SAMPLES,2*N_TRAIN_SAMPLES))
    for i in range(2*N_TRAIN_SAMPLES):
        for j in range(2*N_TRAIN_SAMPLES):
            K_pred[i,j]=kernel_func(A@X[i],A@x[j])
    return K_pred@c

### MAIN ###
sensitivity =
↪  np.empty((len(MU_2_List),NUM_PLOT_POINTS,NUM_PLOT_POINTS))
for MU_2_indx in range(len(MU_2_List)):
    MU_2 = MU_2_List[MU_2_indx]
    # generate 2d-gaussian data (n samples per class) for two
    ↪  classes: x0 and x1
    mean_class_0=np.array([MU_1, MU_2])
    mean_class_1=-mean_class_0
    cov=SCALE_COV*np.array([[1.0, COR],[COR, 1.0]])
    rng = np.random.default_rng()
    x0 = rng.multivariate_normal(mean_class_0, cov,
    ↪  size=N_TRAIN_SAMPLES)
    y0 = np.zeros(N_TRAIN_SAMPLES)
    x1 = rng.multivariate_normal(mean_class_1, cov,
    ↪  size=N_TRAIN_SAMPLES)
    y1 = np.ones(N_TRAIN_SAMPLES)
    x=np.concatenate((x0,x1),axis=0)
    y=np.concatenate((y0,y1),axis=0)
    X=x

    for i_a1 in range(NUM_PLOT_POINTS):
        print ('completion:',i_a1+1,'of',NUM_PLOT_POINTS)
        a1 = (4.0*(i_a1+1))/NUM_PLOT_POINTS
        for i_a2 in range(NUM_PLOT_POINTS):
            a2 = (4.0*(i_a2+1))/NUM_PLOT_POINTS
            A=np.diag([a1,a2])
            pred = kernel_predict(A,x,y,X,kernel)
            A_Perturbed = A + [[DELTA*np.abs(a1),0],[0,0]]
            pred_inc_a1 = kernel_predict(A_Perturbed,x,y,X,kernel)
            A_Perturbed = A + [[0,0],[0,DELTA*np.abs(a2)]]
            pred_inc_a2 = kernel_predict(A_Perturbed,x,y,X,kernel)
            # normalize prediction vectors
```

```
        pred = pred/np.sqrt(np.dot(pred,pred))
        pred_inc_a1 =
        ↪  pred_inc_a1/np.sqrt(np.dot(pred_inc_a1,pred_inc_a1))
        pred_inc_a2 =
        ↪  pred_inc_a2/np.sqrt(np.dot(pred_inc_a2,pred_inc_a2))
        sensitivity[MU_2_indx,i_a1,i_a2] = np.sign( (
        ↪  np.abs(np.dot(pred,pred_inc_a1)-1.0) /
        ↪  (DELTA*np.abs(a1))-np.abs(np.dot(pred,pred_inc_a2)-1.0)
        ↪  / (DELTA*np.abs(a2)) ) * (a1-a2) )

# Plot Sensitivity
xv, yv = np.meshgrid(np.linspace(0, 4, NUM_PLOT_POINTS),
↪  np.linspace(0, 4, NUM_PLOT_POINTS))
fig, (ax1, ax2) = plt.subplots(2)
ax1.pcolormesh(xv, yv, sensitivity[0], vmin = -1, vmax = 1);
ax2.pcolormesh(xv, yv, sensitivity[1], vmin = -1, vmax = 1);
ax1.set_box_aspect(1)
ax2.set_box_aspect(1)
plt.show()
```

