# OpenReview forum: "On the Foundations of Shortcut Learning"
_ICLR.cc/2024/Conference — ICLR 2024 spotlight_

### Official Review · Reviewer_gtPN · 2023-10-28

**Soundness:** 2 fair
**Presentation:** 4 excellent
**Contribution:** 2 fair
**Rating:** 6
**Confidence:** 3

**Summary:**

This work studies the relationship between predictivity and availability of input features, and how a model depends on them. The work starts out with a 2D synthetic scenario where these two dimensions are explicitly characterised. In this scenario, they find that models can rely on more available features, in spite of others being more predictive, and that model depth and nonlinearity increases this effect. The authors further test a synthetic, high-dimensional image dataset and natural images which are manipulated. They find that the number of pixels which correspond to shortcut features, which is viewed as availability of a feature, can lead to the model relying more on it, even when said feature is less predictive. They also provide theoretical results relating to the Neural Tangent Kernel.

**Strengths:**

This work is a creative way of understanding shortcut learning, and I enjoyed reading the paper. The paper clearly defines central concepts and terms, and quantifies them. The experimental analyses are likewise rigorous, plentiful, and well-supported, even though I have important concerns which are listed below. The paper is well-written, mathematically precise, and clear. Overall, this paper will be of interest to the research community.

**Weaknesses:**

The work has several weaknesses, and I have the following concerns:

* The novelty of the work is unclear (see my comment below).
* The synthetic experiment in section 3 is—as much as I enjoyed reading it—highly constructed. It is a good opener for the work, and it draws intuitive, story-supporting conclusions, but I would like to ask the question to which degree the stated conclusions will generalise to high-dimensional datasets in the wild, and generally to which degree these conclusions are generally true. I recognise that this a first step towards this goal, but further evidence or explanation on why these conclusions may be generally true would be useful. Also, it should be considered if section 3 should be given less prominence.
* The naturalistic data experiment in section 7 is most questionable to me. In my view, decreasing the “availability” of the background image by setting its entropy to 0 (black color) is directly biasing the classifier to learn from the remaining available, foreground signal where you know the label is highly predictive. The observed relationship of higher model accuracy is hence non surprise. The same applies to the other scenarios looked at.

**Questions:**

In section 3, the two main parameters that govern availability are $\alpha_i$ and $eta_i$. $\eta_i=0$, so $\alpha_i$ remains, and its ratios are manipulated. Could you please explain your intuition why $\alpha_i$ controls “how easily the feature can be extracted, or leveraged, from inputs”? Why is a feature with a large latent amplitude more readily produced by a neural network? This important premise of your experiment in characterising availability is unclear to me. Any further evidence that makes this clearer would be appreciated.

I don’t fully understand why your definition of reliance is useful. $\hat{y}_\mathcal{M}(z)$ is the “binary classification decision”, i.e. -1 or 1. Then, when taking the expected value over z, I don’t understand why taking a sign difference of z_s and z_c is reasonable.

The novelty of this work relative to related work, which I am not very familiar with, is unclear to me and not discussed in the paper. Could you please delineate your work, and clarify your novel contributions, or point me to appropriate positions in the paper which describe this, if missed?

I would be interested in more complex synthetic scenarios than the one outlined. Did you think about these? How could the robustness of your conclusions in the synthetic scenario be tested?

---

> ### Author Response · Authors · 2023-11-20
> **Re: Official Review by Reviewer gtPN**
>
> Thank you for your questions and feedback!
>
> > ...decreasing the “availability” of the background image by setting its entropy to 0 (black color) is directly biasing the classifier to learn from the remaining available, foreground signal where you know the label is highly predictive.
>
> We want to highlight the fact that our availability manipulation leaves the feature-label statistics (the relationship between background and label, across the dataset) unchanged, and changes only the representation of the background feature that is present within the image. In other words, for a given predictivity setting, the mutual information of the background to label, I(Background; Y), and foreground to label,  I(Foreground; Y), is constant regardless of availabilities. We independently manipulate predictivity and availability with the goal of examining the contributions of these two distinct factors on a model’s feature reliance. The general pattern we observe in the experiment you mention – that decreasing the number of patches of background decreases background preference – is indeed not unexpected, but by systematically manipulating these factors, we can characterize quantitative patterns in the way these contributions interact.
>
> > In section 3, the two main parameters that govern availability are alpha_i and eta_i = 0, so alpha_i remains...explain your intuition why alpha_i controls “how easily the feature can be extracted, or leveraged, from inputs”? ...
>
> With our controlled data setup, we manipulate hypothesized notions of availability, and test whether models are indeed sensitive to them. We hypothesized that $\alpha_i $ (scaling factor) is a notion of availability which pushes around models’ feature use, and find empirically that this is indeed the case (hence the statement).
>
> > I don’t fully understand why your definition of reliance is useful...why taking a sign difference of z_s and z_c is reasonable.
>
> Thank you for this comment. It led us to realize a notational mistake (missing absolute values) in our reliance definition in Section 3, which we’ve now updated (all figures and results remain correct and are unchanged). Please refer to the updated equation in the updated paper PDF. To answer your question, our "reliance" measure quantifies the extent to which a model uses feature $z_s$ as the basis for its classification decisions – intuitively, the extent to which the model decisions match those which would be generated by a classifier which relied exclusively on $z_s$ versus by a classifier which relied exclusively on $z_c$. To capture this notion, we take a difference between a template match with each of these classifiers. By construction of the Gaussian dataset, a value’s sign is its feature label.
>
> > The novelty of this work relative to related work, which I am not very familiar with, is unclear to me and not discussed in the paper. Could you please delineate your work, and clarify your novel contributions, or point me to appropriate positions in the paper which describe this, if missed?
>
> We summarize our contributions at the end of the Introduction, and situate these contributions with respect to the literature in both the Intro and Related Work sections – see especially the paragraph beginning, “The literature examining feature use…”. Briefly: we explicitly introduce “availability’’ to refer to the factors which influence the likelihood that a model will use a feature more than a purely statistical account would predict. We systematically manipulate both predictivity and availability and give a quantitative account of their interaction, characterizing conditions giving rise to shortcut learning. In experiments with controlled and naturalistic data, we identify types of availability that drive model behavior. We find that availability-driven shortcut bias is greater for nonlinear than linear models, and that model depth amplifies bias. We also provide a novel theoretical analysis, consistent with the empirical results, indicating the inevitability of a shortcut bias for a ReLU network compared to a linear one. Further, we show that standard vision architectures used in practice are sensitive to availability manipulations. Please let us know if any further clarification would be helpful!
>
> > I would be interested in more complex synthetic scenarios than the one outlined. Did you think about these?
>
> Yes, definitely. We focused on a setting in which we could explicitly control notions of availability which it’s reasonable to hypothesize could drive feature reliance in real models, and considered a variety of types of data, from vector inputs to synthetic images to naturalistic images. We found that across several different notions of availability, models exhibited a shortcut bias, which is not what a purely statistical account would predict. We are very interested in future work to surface even more types of availability that drive the behavior of models used in practice.

---

> > ### Comment · Reviewer_gtPN · 2023-11-22
> > **Thank you**
> >
> > Thank you for your updates to the manuscript and the useful explanations. I agree with and understand all, but I still don't fully understand the connection between $\alpha_i$ and availability, and why this holds in general, not just the presented experiments. -- In your comment, I don't understand what "hypothesized notions of availability" means.

---

### Official Review · Reviewer_G1nt · 2023-10-30

**Soundness:** 3 good
**Presentation:** 2 fair
**Contribution:** 3 good
**Rating:** 8
**Confidence:** 4

**Summary:**

This paper presents a systematic study on the trade-off between predictivity and availability in deep learning, proposing a theoretical foundation for shortcut learning in neural networks. The key contributions are:
1.The paper introduces quantitative notions of predictivity and availability using a generative model to synthesize classification datasets where two latent features vary in terms of these attributes. A measure of "shortcut bias" is proposed to quantify a model's over-reliance on more available but less predictive features.
2.Through controlled experiments, it is shown that nonlinear models exhibit greater shortcut bias compared to linear models, and model depth amplifies this effect.
3.A theoretical analysis based on Neural Tangent Kernels proves the inevitability of availability bias for nonlinear architectures like ReLU networks, while linear networks are unbiased.
4.Experiments on natural image datasets demonstrate that widely used vision models are sensitive to availability, not just predictivity, of non-core features. Explicit availability manipulations of images are shown to alter models' reliance on different features.
Taken together, the empirical and theoretical findings reveal shortcut learning as an inherent characteristic of nonlinear deep models that needs to be studied systematically. The framework presented lays the groundwork for further investigating architectural choices, dataset factors and methods to mitigate shortcut bias.
5.Overall, this is a thorough, well-executed study that makes important theoretical and practical contributions towards understanding shortcut learning in deep neural networks. The paper is clearly written and the empirical methodology is sound. The theorems and proofs rigorously analyze the interplay between predictive value and feature availability. This work provides key insights into model failures related to over-reliance on spuriously predictive features, and tools to improve model robustness.

**Strengths:**

1.The proposed generative framework for synthesizing datasets with controllable levels of predictivity and availability is creative and enables controlled experimentation.
2.The empirical methodology is thorough and sound. The datasets, models, and evaluation metrics are carefully designed. Results are reported over multiple random seeds to ensure significance.
3.Shortcut learning is a pivotal concept in deep learning with implications for generalization and fairness. This work significantly advances our understanding of why models fail in this manner.
In summary, this is a paper of exceptional quality and scientific merit that offers significant theoretical and practical value to the field. The novel concepts, thorough empirics, and rigorous theory set a new standard and open up many promising research directions.

**Weaknesses:**

This is an good paper overall, but a few minor weaknesses could be addressed to further improve it:

- The theoretical analysis makes approximations to obtain tractability (small covariance, quadratic approximation of ReLU kernel). It would strengthen the analysis to discuss the impact of these approximations. Are the core insights still valid without them?

- The proposed notion of availability is intuitively reasonable but remains a hypothesis. Additional ablation studies that directly validate the choice of manipulations affecting availability could make this more concrete.

- The measures of reliance and shortcut bias, though well-motivated, are indirectly quantified through alignment with idealized classifiers. More analysis could be provided to justify these metrics over alternatives.

- There is scope for further investigating architectural manipulations that could mitigate shortcut bias, beyond just model depth and activation function. This could lead to practical guidelines.

Overall these are minor limitations that do not diminish the quality of the work. The paper thoroughly delivers on its core promises. Addressing the above points where possible would make it even stronger. The work clearly advances our understanding of an important problem and provides a solid foundation for reducing shortcut reliance in deep learning models.

**Questions:**

Here are some questions and suggestions to further improve the paper:

- The quadratic approximation for the ReLU kernel greatly simplified the theoretical analysis. Could you provide some empirical verification that this does not alter the core findings? For example, compare the availability bias for the true ReLU kernel versus the quadratic approximation.

- Have you experimented with any architectural manipulations beyond depth and activation functions that could potentially reduce shortcut bias? Things like skip connections, normalization layers etc. Exploring this could provide practical guidelines.

- For the image experiments, it would be interesting to also show the impact of availability manipulations on a model pretrained on Imagenet. Does pretraining affect sensitivity to shortcuts?

- The measures of reliance and bias involve probing a model in latent space. For vision experiments, could you provide some visualizations of model decisions before/after availability manipulations to build intuition?

- The notion of availability is central but still somewhat conceptual. Are there any additional experiments you could do to further validate the manipulations proposed to affect availability?

I hope these suggestions help further refine the work. The key results seem solid and demonstrate the importance of availability bias. Some additional experiments and discussion along the lines above could make it even more convincing and applicable across domains. I look forward to the authors' response.

**Details Of Ethics Concerns:**

none.

---

> ### Author Response · Authors · 2023-11-20
> **Re: Official Review by Reviewer G1nt**
>
> Thank you for these questions and comments!
>
> > The theoretical analysis makes approximations to obtain tractability (small covariance, quadratic approximation of ReLU kernel). It would strengthen the analysis to discuss the impact of these approximations. Are the core insights still valid without them?...The quadratic approximation for the ReLU kernel greatly simplified the theoretical analysis. Could you provide some empirical verification that this does not alter the core findings? For example, compare the availability bias for the true ReLU kernel versus the quadratic approximation.
>
> Thank you for raising this point. As mentioned in the Global response above, based on your comment, we ran some simulations to compare the behavior of the ReLU kernel and its quadratic approximation. The simulations work with a real (as opposed to asymptotically small) covariance matrix. The results show a good match between the two kernels, as well as with the purely analytical predictions in Figure 5 of the main paper. We added these results and the Python code used to generate them to Appendix I of the updated Supplementary Material PDF. Due to the limited time we had for the rebuttal, we were able to do a minimal set of experiments to address your concern. We will expand and add more results to this section in the camera-ready version.
>
> > The proposed notion of availability is intuitively reasonable but remains a hypothesis. Additional ablation studies that directly validate the choice of manipulations affecting availability could make this more concrete. The measures of reliance and shortcut bias, though well-motivated, are indirectly quantified through alignment with idealized classifiers. More analysis could be provided to justify these metrics over alternatives.
>
> Just to make sure we’re understanding: we do directly manipulate factors which we hypothesize affect availability. We observe an effect of these factors on shortcut learning, thereby obtaining support for our hypothesis. If the worry is that we don’t rule out other factors that affect availability, we agree and would not want to claim that we’ve identified all such factors.
>
> > Have you experimented with any architectural manipulations beyond depth and activation functions that could potentially reduce shortcut bias?
>
> Things like skip connections, normalization layers etc. Exploring this could provide practical guidelines.
> We are also interested in these questions and plan to explore them in future work.
>
> > For the image experiments, it would be interesting to also show the impact of availability manipulations on a model pretrained on Imagenet. Does pretraining affect sensitivity to shortcuts?
>
> Good question! In SI Figure B.9, we found that ImageNet-pretrained models are still sensitive to availability manipulations, though there is a difference from models trained from scratch.
>
> > The measures of reliance and bias involve probing a model in latent space. For vision experiments, could you provide some visualizations of model decisions before/after availability manipulations to build intuition?
>
> In experiments shown in Figures 6 and B.9, we had found that the accuracy of vision models at classifying Waterbirds images by the core feature (bird) is influenced by non-core feature (background) availability. In Figure B.10, for a particular background availability manipulation, we compare attribution maps for a ResNet18 when trained on one level of availability manipulation versus another, and see an expected difference in focal point.

---

### Official Review · Reviewer_KL3Q · 2023-11-01

**Soundness:** 3 good
**Presentation:** 3 good
**Contribution:** 3 good
**Rating:** 8
**Confidence:** 3

**Summary:**

The paper studies shortcut learning, i.e., why networks use to learn shortcut/spurious features over intended semantic features. The authors suggest that the network prefers a feature that is more available to quantify shortcut bias in terms of how a learned classifier deviates from an optimal classifier in its feature reliance. The authors study the neural network preference toward shortcut features using synthetic datasets with varying predictability and availability of the shortcut features. They empirically observed that networks prefer to learn shortcut features when they are more available, and ReLU is biased toward shortcuts. They also theoretically show using the NTK that linear networks are less biased towards the shortcut compared to the ReLU networks.

**Strengths:**

* The paper is well-written and easy to follow. The problem of shortcut learning, which is not extensively studied, is important to understand.

* The paper presents interesting insights into neural networks’ preference toward shortcuts. The paper studies shortcut learning empirically using controlled datasets and theoretically using the NTK.

* The derivation for the bias of linear and ReLU networks using NTK would be helpful for future work in this domain.

**Weaknesses:**

* The main observation that the model depth and non-linearity increase bias towards the shortcut features is intuitive. Both depth and non-linearity allows the model to learn a rich representation.

* Theoretical analysis using NTK is problematic as they don't learn feature and thus cannot necessarily explain model's preference towards the shortcut feature.

* Only vision tasks are explored in the paper, it is not clear if the observations will hold true for other domains.

* It would be interesting to see experiments with the vision transformers.

**Questions:**

* Do you think the observations will be similar for other non-linearities?

---

> ### Author Response · Authors · 2023-11-20
> **Re: Official Review by by Reviewer KL3Q**
>
> Thank you for your questions and comments!
>
> > The main observation that the model depth and non-linearity increase bias towards the shortcut features is intuitive. Both depth and non-linearity allows the model to learn a rich representation.
>
> On the other hand, one might predict that the rich representation would enable models to go *beyond* shortcut features (rely on them less). Of course, this is not what happens, but we believe a few intuitive accounts are possible here, making the empirical verification worthwhile.
>
> > Theoretical analysis using NTK is problematic as they don't learn feature and thus cannot necessarily explain model's preference towards the shortcut feature.
>
> Although we agree that learning has a big role in creating shortcut features, there is also opportunity/potential for shortcut features inherent in the architecture itself.  Our theory focuses on the latter via the NTK.
>
> > It would be interesting to see experiments with the vision transformers.
>
> We agree that in future work, it will be interesting to study shortcut learning in domains beyond the ones we have focused on here, and to add experiments with ViTs.
>
> > Do you think the observations will be similar for other non-linearities?
>
> We do! In Figures 3B and B.1, we find that, empirically, models with a Tanh activation function induce a larger shortcut bias than a linear counterpart.

---

> > ### Comment · Reviewer_KL3Q · 2023-11-22
> > **Re:**
> >
> > Thanks for addressing my questions/concerns! After reading the rebuttal to all the reviews, I feel more confident that this paper would be a good asset to the research community.

---

### Official Review · Reviewer_mfV7 · 2023-11-01

**Soundness:** 3 good
**Presentation:** 2 fair
**Contribution:** 3 good
**Rating:** 6
**Confidence:** 3

**Summary:**

The paper investigates the learning of shortcut / spurious features. Mainly, the paper looks at the interplay between predictability and availability (that defines how easy it is for a model to extract a feature). First, experiments on simple, synthetic data, generated from two variables for which we can control the predictability and availability are shown. A notion of shortcut bias is defined as the additional reliance on spurious features for a learned model, compared to an optimal model. It is shown that availability determines the learning of spurious features, even when the predictivity of spurious features is lower than that of the core features. Non-linearities are shown to induce more bias for shortcuts. Theoretical analysis shows that linear networks are not biased to feature availability while ReLU networks are. Experiments on image datasets show that they are biased for background and object size.

**Strengths:**

- S1. The paper deals with an important aspect of understanding neural networks, the bias for learning shortcuts.

- S2. Using the notion of shortcut bias, based on the reliance of an optimal predictor is a good idea.

- S3. Analysing based on a notion of availability, that can be computed produces some good observations.

- S4. Interesting to see that theoretically, ReLU networks are more biased than linear networks.

**Weaknesses:**

- W1. The notion of availability is very closely connected with the concept of simplicity of neural networks. The simplicity bias has been pointed out as a cause of non-robust learning. The paper gives multiple references (e.g. Shah et al., 2020, etc.) for works dealing with simplicity bias, but they are not discussed in detail. The authors must clarify what is the difference between the notion of availability presented in this work, and the simplicity bias previously introduced. What new observations does the proposed framework bring?


- W2.The experiments are not that strong. The image datasets do not seem to bring any interesting observations. It is already known that the background influences the prediction to a high degree. It is expected that alterating the background will improve the reliance on the core features. All methods that use these datasets, try to learn how to not rely on the background. There doesn’t seem to be any novel observation in this regard.

- W3. For image datasets: “a Bayes optimal classifier is not comparably sensitive to the predictivity of the non-core features” How do we know this? What experiments give this conclusion?

- W3.2 Also, how is the Bayes optimal classifier created in the case of real image datasets (WaterBirds, CelebA)?

- W4. The paper defined a shortcut as a feature that is more available, but less predictive. Differently, a shortcut is usually defined by good predictivity in distribution, but poor predictivity in some other distributions (OOD).

**Questions:**

Could the curves given by making an intervention on the background, be use to benchmark the robustness of different method? E.g. more robust models would have curves that are less sensitive to interventions on the background. Whould this offer any additional insight as opposed to comparing the accuracy of the model on balanced data, without foreground-background spurious correlations?

---

> ### Author Response · Authors · 2023-11-20
> **Re: Official Review by by Reviewer mfV7**
>
> Thank you for your questions and feedback!
>
> > W1. The notion of availability is very closely connected with the concept of simplicity of neural networks. The simplicity bias has been pointed out as a cause of non-robust learning. The paper gives multiple references (e.g. Shah et al., 2020, etc.) for works dealing with simplicity bias, but they are not discussed in detail. The authors must clarify what is the difference between the notion of availability presented in this work, and the simplicity bias previously introduced. What new observations does the proposed framework bring?
>
> We pointed to Shah et al. (2020), Hermann & Lampinen (2020), and related papers as including motivating examples of cases in which a network may prefer one feature over another even when the two are equally predictive. Shah et al. specifically characterized features as simple or complex in terms of their corresponding decision boundary (e.g. linear or piecewise), and found that models will learn the linear decision boundary, and can ignore complex features. Our paper is different from this earlier work in several ways:
> * Our study of availability is not restricted to cases where features are linearly vs nonlinearly related to task labels. For example, we study cases in which both features are linearly related to labels, but nonetheless one is more available than the other.
> * Unlike the prior studies, we study how models trade off availability with predictivity, which is critical to understanding the conditions under which shortcut learning is likely to occur.
> * We study how feature reliance changes as a function of (continuous) degree of availability, not just model preference given features which differ in kind.
>
> > W3. For image datasets: “a Bayes optimal classifier is not comparably sensitive to the predictivity of the non-core features” How do we know this? What experiments give this conclusion? W3.2 Also, how is the Bayes optimal classifier created in the case of real image datasets (WaterBirds, CelebA)?
>
> The optimal classifier is based on the assumption of ground-truth knowledge of the true bird class and the true background type. As such, it represents an upper bound on accuracy. However, our point in including the Bayes optimal classifier was to examine the small modulations in accuracy that are due to non-core feature predictivity.
>
> > Could the curves given by making an intervention on the background, be use to benchmark the robustness of different method? E.g. more robust models would have curves that are less sensitive to interventions on the background. Whould this offer any additional insight as opposed to comparing the accuracy of the model on balanced data, without foreground-background spurious correlations?
>
> That’s a good question which would be interesting to study in future work!

---

### Official Review · Reviewer_QCsi · 2023-11-05

**Soundness:** 3 good
**Presentation:** 3 good
**Contribution:** 3 good
**Rating:** 6
**Confidence:** 2

**Summary:**

This paper studies why shortcut learning happens. It focuses on two characteristics that input features may have: availability and predictivity. Availability refers to how frequently that type of feature is available in the data, while predictivity means how useful it is in predicting the target. Shortcuts are described as overly relying on available features that are less predictive. Based on this interpretation of shortcuts, the paper experimentally shows on synthetic and natural image datasets that availability explains shortcut learning where predictivity cannot. These findings are supplemented with a theoretical analysis that concludes that adding a single hidden layer already biases the model to rely on shortcuts.

**Strengths:**

The paper is well-written and the main message that availability is a key driver to shortcut learning is convincingly conveyed.

Both empirical support and theoretical support are provided to underline the effect of availability on shortcut learning.

**Weaknesses:**

The theoretical analysis is limited to a single hidden layer MLP which does not reflect the type of architectures used in practice. It is difficult to conclude whether this also holds for other types of architectures like Transformers or CNNs.

Experiments are limited to supervised classification settings. Self-supervised training or other tasks like object detection would be interesting to consider in the context of shortcut learning.

**Questions:**

The paper points out that availability can explain (some) occurrences of shortcut learning. The availability can be determined by looking at the training data distribution, but could one also identify these shortcuts stemming from availability by directly looking at the train neural network weights? For example, if multiple filters in the same CNN layer share the same pattern?

---

> ### Author Response · Authors · 2023-11-20
> **Re: Official Review by Reviewer QCsi**
>
> Thank you for your questions and comments!
>
> > The theoretical analysis is limited to a single hidden layer MLP which does not reflect the type of architectures used in practice. It is difficult to conclude whether this also holds for other types of architectures like Transformers or CNNs.
>
> We agree that our theoretical setup uses a simple architecture, but a precise analysis of the bias even in this setting is highly non-trivial (please check the appendix for a detailed coverage). Moreover, our work is the first that presents any theoretical analysis of the predictivity/availability tradeoff, for which we had to develop a novel proof strategy (as opposed to an incremental refinement to existing results). Our analysis delivers a precise characterization of availability and predictivity which we believe to be a novel contribution.
>
> > Experiments are limited to supervised classification settings. Self-supervised training or other tasks like object detection would be interesting to consider in the context of shortcut learning.
>
> We agree that studying the interaction of availability and predictivity in these other settings would indeed be very interesting! For example, it’s been observed that self-supervised vision models are still texture-biased (e.g., Hermann et al. 2020), though the statistics of ImageNet are unknown and so controlled study is needed. Nevertheless, as per the response above, we believe the analysis in the supervised setting already requires a good deal of non-trivial work; we look forward to expanding to other settings in future work.
>
> > The paper points out that availability can explain (some) occurrences of shortcut learning. The availability can be determined by looking at the training data distribution, but could one also identify these shortcuts stemming from availability by directly looking at the train neural network weights? For example, if multiple filters in the same CNN layer share the same pattern?
>
> Your suggestion is clever and a very sensible idea to try. The redundancy you describe is exactly what we might expect to see for, say, texture patches. It would be a bit challenging to quantify “share the same pattern”, but we agree this idea warrants further consideration in future work.

---

### Author Response · Authors · 2023-11-20
**Global response**

We thank the reviewers for their very thoughtful feedback, and reply to individual points inline below. We have also uploaded updated versions of the paper and Supplementary Material, adding an analysis to the Appendix in response to a suggestion by Reviewer G1nt, and updating a definition in response to a question by Reviewer gtPN, as described inline below. In the new analysis, we empirically analyze the availability/predictivity tradeoff to understand the effect of the approximations used in the theory on the actual results. Specifically, based on the suggestion by Reviewer G1nt, we ran simulations to compare the behavior of the ReLU kernel and its quadratic approximation. The simulations work with a real (as opposed to asymptotically small) covariance matrix. The results show a good match between the two kernels, as well as with the purely analytical predictions in Figure 5 of the main paper. We have added these results and the Python code used to generate them to Appendix I (updated SI PDF).

---

### Meta-Review · Area_Chair_cMHJ · 2023-12-07

**Metareview:**

This paper focuses on the foundation theory of shortcut learning. The authors define quantitative measures of predictability and availability metrics to characterize the shortcut bias. Extensive experiments are conducted to demonstrate that availability explains shortcut learning where predictivity cannot be on both synthetic and natural image datasets. Furthermore, the authors show that the shortcut bias is inevitable for nonlinear neural networks by using the Neural Tangent Kernels techniques. Overall, this work provides a new understanding of shortcut learning.

Strengths:

(1)   Shortcut learning is an important and challenging problem and has many practical applications. The paper deals with an important aspect of understanding shortcuts.

(2)   The paper is well-written and the idea is easy to follow. Both the experiments and theoretical analysis are extensive to support their findings.

(3)   The findings in this work are interesting, and provide a new comprehensive understanding of shortcut learning for nonlinear neural networks.

Weaknesses:

(1)   The theoretical analysis is limited to a two-layer network which does not reflect the type of architectures used in practice. Nowadays, there exist several works focus on the theory of multi-layer deep neural networks (CNNs and Transformers). The authors mention the restrictions of the theory analysis. Furthermore, we recommend the authors give more discussions on the challenges when we extend the analysis to the deeper neural network.

After the authors' response and discussion with reviewers, all the concerns are well-solved. All the reviewers agree to accept this work. Thus, I recommend acceptance.

**Justification For Why Not Higher Score:**

The theory mainly focuses on a two-layer deep neural network, which limits its practical applications.

**Justification For Why Not Lower Score:**

N/A

---

### Decision · Program_Chairs · 2024-01-16

Accept (spotlight)